# Prospective Learning: Learning for a Dynamic Future

**Ashwin De Silva**[*,1]  **Rahul Ramesh**[*,2]  **Rubing Yang**[*,2]

**Siyu Yu**[1]  **Joshua T. Vogelstein**[†,1]  **Pratik Chaudhari**[†,2]

[*,†] **Equal Contribution**
Email: {ldesilv2, syu80, jovo}@jhu.edu, {rahulram, rubingy, pratikac}@upenn.edu

## Abstract

In real-world applications, the distribution of the data, and our goals, evolve over time. The prevailing theoretical framework for studying machine learning, namely probably approximately correct (PAC) learning, largely ignores time. As a consequence, existing strategies to address the dynamic nature of data and goals exhibit poor real-world performance. This paper develops a theoretical framework called "Prospective Learning" that is tailored for situations when the optimal hypothesis changes over time. In PAC learning, empirical risk minimization (ERM) is known to be consistent. We develop a learner called Prospective ERM, which returns a sequence of predictors that make predictions on future data. We prove that the risk of prospective ERM converges to the Bayes risk under certain assumptions on the stochastic process generating the data. Prospective ERM, roughly speaking, incorporates time as an input in addition to the data. We show that standard ERM as done in PAC learning, without incorporating time, can result in failure to learn when distributions are dynamic. Numerical experiments illustrate that prospective ERM can learn synthetic and visual recognition problems constructed from MNIST and CIFAR-10. Code at https://github.com/neurodata/prolearn.

## 1   Introduction

All learning is for the future. Learning involves updating decision rules or policies, based on past experiences, to improve future performance. Probably approximately correct (PAC) learning has been extremely useful to develop algorithms that minimize the risk—typically defined as the expected loss—on unseen samples under certain assumptions. The assumption, that samples are independent and identically distributed (IID) within the training dataset and at test time, has served us well. But it is neither testable nor believed to be true in practice. The future is always different from the past: both distributions of data and goals of the learner may change over time. Moreover, those changes may cause the optimal hypothesis to change over time as well. There are numerous mathematical and empirical approaches that have been developed to address this issue, e.g., techniques for being invariant to [1], or adapting to, distribution shift [2], modeling the future as a different task, etc. But we lack a first-principles framework to address problems where data distributions and goals may change over time in such a way that the optimal hypothesis is time-dependent. And as a consequence, machine learning-based AI today is brittle to changes in distribution and goals.

This paper develops a theoretical framework called "Prospective Learning" (PL). Instead of data arising from an unknown probability distribution like in PAC learning, prospective learning assumes that data comes from an unknown stochastic process, that the loss considers the future, and that the optimal hypothesis may change over time. A prospective learner uses samples received up to some time $t \in \mathbb{N}$ to output an infinite sequence of predictors, which it uses for making predictions on data at all future times $t' > t$. We discuss how prospective learning is related to existing problem formulations in the literature in Section 3 and Appendix A.

**Why should one care about prospective learning?**   Imagine a deployed machine learning system. The designer of this system desires to optimize—not the risk upon the past training data, or the risk

on the immediate future data—but the risk on all data that the model will make predictions upon in the future. As data evolves, e.g., due to changing trends and preferences of the users, the optimal hypothesis to make predictions also changes. Time is the critical piece of information if the system designer is to achieve their goals. Both in the sense of how far back in time a particular datum was recorded, and in the sense of how far ahead in the future this system will be used to make predictions. The designer must take time into account to avoid retraining the model periodically, *ad infinitum*.

Biology is also rich with examples where systems seem to behave prospectively. The principle of allostasis, for example, states that regulatory processes of living things anticipate the needs of the organism and prepare to satisfy these needs before, rather than after, they arise [3]. For example, mitochondria increase their energy production to anticipate the demands of muscles [4], neural circuits anticipate changes in sensory stimuli and the task (i.e., predictive coding [5]), and individual organisms optimize their actions with respect to anticipated changes in their environments [6, 7]. These regulatory principles were learned early in evolutionary time so they must be important. In short, the world—including our internal drives—changes all the time, and learning systems must anticipate (that is, prospect) these changes to thrive.

**Contributions**

- Section 2 defines Prospective Learning (PL) as an approach to address problems where the optimal hypothesis may evolve over time (due to shifts in distributions and/or goals of the learner). It also provides illustrative examples of PL.
- Section 3 and Appendix A put prospective learning in context relative to existing ideas in the literature to address changes in the data distribution.
- Section 4 takes steps towards a theoretical foundation for prospective learning. We define strongly learnability (i.e., there exists a prospective learner whose risk is arbitrarily close to the Bayes optimal learner) and weakly learnability (i.e., there exists a prospective learner whose risk is better than chance) [8]. Empirical risk minimization (ERM) without incorporating time, can result in failure to strongly, or even weakly, learn prospectively [9].
- Section 5 introduces prospective empirical risk minimization, and proves that it can learn prospectively under certain assumptions on the stochastic process and loss.
- Section 6 demonstrates that our prospective learners can prospectively learn several canonical problems constructed using synthetic, MNIST [10] and CIFAR-10 [11] data. In contrast, a number of existing algorithms, including ERM, online and continual learning algorithms, fail. Appendix H demonstrates that current large language models, which use Transformer-based architectures trained using auto-regressive losses, fail to learn prospectively.

## 2    A definition of prospective learning

A prospective learner minimizes the expected cumulative risk of the future using past data. Such a learner is defined by the following key ingredients (see Fig. 1 (left) for schematic illustration).

**Data.** Let the input and output at time $t$ be denoted by $x_t \in \mathcal{X}$ and $y_t \in \mathcal{Y}$ respectively. Let $z_t = (x_t, y_t)$. We will model the data as a stochastic process $Z \equiv (Z_t)_{t \in \mathbb{N}}$ defined on an appropriate probability space $(\Omega, \mathcal{F}, \mathbb{P})$. At time $t \in \mathbb{N}$, denote past data by $z_{\leq t} \equiv (z_1, \ldots, z_t)$ and future data by $z_{>t} \equiv (z_{t+1}, \ldots)$. We will find it useful to distinguish between the realization of the data, denoted by $z_{\leq t}$, and the corresponding random variable, $Z_{\leq t}$.

**Hypothesis class.** At each time $t$, a prospective learner selects an infinite sequence $h \equiv (h_1, \ldots, h_t, h_{t+1}, \ldots)$ which it uses to make predictions on data at any time in the future. Each element of this sequence $h_t : \mathcal{X} \mapsto \mathcal{Y}$ and therefore $h_t \in \mathcal{Y}^{\mathcal{X}}$.[1] The hypothesis class $\mathcal{H}$ is the space of such hypotheses, $h \in \mathcal{H} \subseteq (\mathcal{Y}^{\mathcal{X}})^{\mathbb{N}}$.[2] We will again use the shorthand $h_{\leq t} \equiv (h_1, \ldots, h_t)$. We will sometimes talk about a "time-agnostic hypothesis" which will refer to a hypothesis such that $h_t = h_{t'}$ for all $t, t' \in \mathbb{N}$. Observe that this makes our setup different from the standard setup in PAC learning where the learner selects a single hypothesis in $\mathcal{Y}^{\mathcal{X}}$. One could also think of prospective learning as using a single time-varying hypothesis $h : \mathbb{N} \times \mathcal{X} \mapsto \mathcal{Y}$, i.e., the hypothesis takes both time and the datum as input to make a prediction.

**Learner.** A prospective learner is a map from the data received up to time $t$, to a hypothesis that makes predictions on the data over all time (past and future): $(\mathcal{X} \times \mathcal{Y})^t \to (\mathcal{Y}^{\mathcal{X}})^{\mathbb{N}}$. The learner gives

---

[1]We will use some non-standard notation in this paper. In particular, a hypothesis $h$ will always refer to sequence of predictors $h \equiv (h_1, \ldots, h_t, h_{t+1}, \ldots)$. This helps us avoid excessively verbose mathematical expressions.

[2]When we say that "learner selects a hypothesis" in the sequel, it will always mean that the learner selects an infinite sequence from within the hypothesis class $\mathcal{H}$.

as output a hypothesis $h(z_{\leq t}) \in \mathcal{H}$. Unlike a PAC learner, a prospective learner can make different kinds of predictions at different times. This is a crucial property of prospective learning. In other words, after receiving data up to time $t$, the hypothesis selected by the prospective learner can predict on samples at any future time $t' > t$.

**Prospective loss and risk.** The future loss incurred by a hypothesis $h$ is

$$\bar{\ell}_t(h, Z) = \limsup_{\tau \to \infty} \frac{1}{\tau} \sum_{s=t+1}^{t+\tau} \ell(s, h_s(X_s), Y_s), \tag{1}$$

where $\ell : \mathbb{N} \times \mathcal{Y} \times \mathcal{Y} \mapsto [0, 1]$ is a bounded loss function.[3] Prospective risk at time $t$ is the expected future loss [4]

$$R_t(h) = \mathbb{E}\left[\bar{\ell}_t(h, Z) \mid z_{\leq t}\right] = \int \bar{\ell}_t(h, Z) \, d\mathbb{P}_{Z|z_{\leq t}}, \tag{2}$$

where we assume that $h$ is a random variable and $h \in \sigma(Z_{\leq t})$ where $\sigma(\cdot)$ denotes the filtration (an increasing sequence of sigma algebras) of the stochastic process $Z$. We have used the shorthand $\mathbb{E}[Y \mid x]$ for $\mathbb{E}[Y \mid X = x]$. Observe that we have conditioned the prospective risk of the hypothesis $h$ upon the realized data $z_{\leq t}$. We can take an expectation over the realized data, to obtain the expected prospective risk

$$\mathbb{E}\left[R_t(h)\right] = \int R_t(h) \, d\mathbb{P}_{Z_{\leq t}}.$$

**Prospective Bayes risk** is the minimum risk achievable by any hypothesis. In PAC learning, it is a constant that depends upon the (fixed) true distribution of the data and the risk function. In prospective learning, the optimal hypothesis can predict differently at different times. We therefore define the prospective Bayes risk at a time $t$ as

$$R_t^* = \inf_{h \in \sigma(Z_{\leq t})} R_t(h), \tag{3}$$

which is the minimum achievable prospective risk by any learner that observes data $z_{\leq t}$. We define the Bayes optimal learner as any learner that achieves a Bayes optimal risk at every time $t \in \mathbb{N}$. In certain contexts, one might be interested in the limiting prospective Bayes risk as $t \to \infty$.

## 2.1 Different prospective learning scenarios with illustrative examples

We next discuss four prospective learning scenarios that are relevant to increasingly more general classes of stochastic processes. Our goal is to illustrate, using examples, how the definitions developed in the previous section capture these scenarios. We will assume that for all times $t$ we have $X_t = 1$, $Y_t \in \{0, 1\}$. We will also focus on the time-invariant zero-one loss $\ell(t, \hat{y}, y) = \delta(\hat{y} \neq y)$ for all $t$, here $\delta$ is the Dirac delta function. Fig. 1 shows example realizations of the data for each scenario.

**Scenario 1** (**Data is independent and identically distributed**). Formally, this consists of stochastic processes where $\mathbb{P}_{Z_{t'}|Z_{\leq t}} = \mathbb{P}_{Z_t}$ for all $t, t' \in \mathbb{N}$. As an example, consider $Y_t \sim \text{Bernoulli}(p)$ for some unknown parameter $p \in [0, 1]$. Prospective Bayes risk is equal to $\min(p, 1 - p)$ in this case. A time-agnostic hypothesis, for example one that thresholds the maximum likelihood estimator (MLE) of the Bernoulli probability, converges to the limiting prospective Bayes risk.[5]

**Scenario 2** (**Data is independent but not identically distributed**). This consists of stochastic processes where $\mathbb{P}_{Z_t|Z_{\leq t}} = \mathbb{P}_{Z_t}$ for all $t \in \mathbb{N}$. Consider $Y_t \sim \text{Bernoulli}(p)$ if $t$ is odd, and $Y_t \sim \text{Bernoulli}(1 - p)$ if $t$ is even, i.e., data is drawn from two different distributions at alternate times. Prospective Bayes risk is again equal to $\min(1 - p, p)$ in this case. A time-agnostic hypothesis can only perform at chance level. But a prospective learner, for example one that selects a hypothesis that alternates between two predictors at even and odd times, can converge to prospective Bayes optimal risk. We can also construct variants, e.g., when the relationship between the Bernoulli probabilities are not known (Variant 1 in Fig. 1), or when the learner does not know that the data distribution changes at every time step (Variant 2 in Fig. 1 where we implemented a generalized likelihood ratio

---

[3]The limsup is guaranteed to exist if $\ell$ is bounded. If the series converges, we can use lim instead.

[4]There are many real world scenarios where expected future loss may not be sufficient for good performance, e.g., for portfolio managements or inference by biological learners who optimize for a balance between value and risk. Moreover, the risk functional could, in general, also change over time. In this paper, we will focus only on the expected future loss.

[5]We show an interesting observation in Appendix B.1: if the prior of a Bayesian learner is different from the true Bernoulli probability, then prospective learning can improve upon the maximum *a posteriori* estimator.

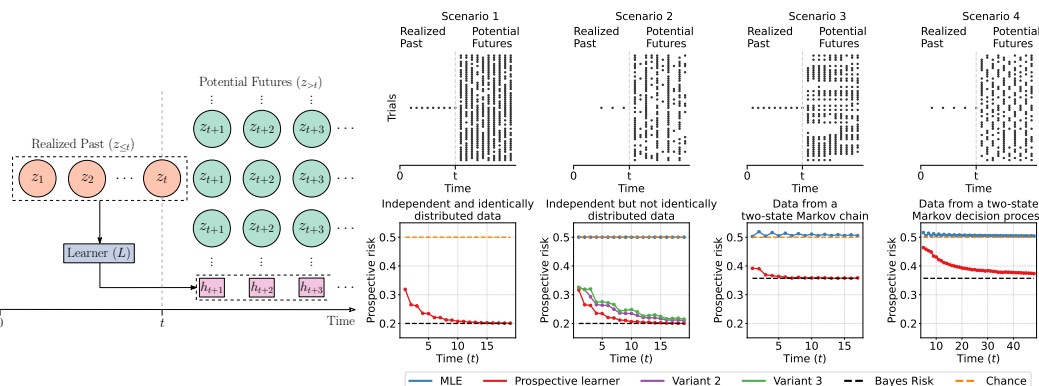

**Figure 1:** A schematic for prospective learning (left) and realizations of the examples for the four scenarios (top right); dots denote 1s and empty spaces denote 0s for $Y_t \in \{0, 1\}$ with $X_t = 1$ for all times $t$. Prospective risk of learners at different times is shown in the bottom panels and discussed in Section 2.1. **Scenario 1:** For Bernoulli probability $p = 0.2$, the maximum-likelihood estimator (MLE) in blue uses a time-agnostic hypothesis $h_t(X_t) = \mathbf{1}(\hat{p}_t > 0.5)$ where $\hat{p}_t = t^{-1} \sum_{s=1}^{t} y_s$, ties at $\hat{p}_t = 0.5$ are broken randomly. The risk of this learner converges to the Bayes risk. **Scenario 2:** For Bernoulli probability $p = 0.2$, the MLE estimator (blue) performs at chance levels. A prospective learner (red) that alternates between two predictors at even and odd times converges to Bayes risk. Variants of this learner that use less information from the stochastic process (purple does not know that the data distributions at even and odd times are tied, green does not know that the distribution shifts at every time-step) also converge to Bayes risk, but more slowly. **Scenario 3:** For $\theta = 0.1$ and $\gamma = 0.9$ in the discounted prospective risk, the MLE estimator (blue) again performs at chance levels. A prospective learner that computes an estimate of the transition probability of the two-state Markov chain to estimate $\mathbb{P}(Y_{t'} \mid y_t)$ for future times $t' > t$ converges to Bayes risk. **Scenario 4:** For $\theta_0 = \theta_1 = 0.1$, the MLE estimator (blue) performs at chance levels. A prospective learner that uses a variant of Q-learning (described in the text and Appendix B.3) converges to the prospective Bayes risk.

test to determine whether the distribution changes). The risk of these variants also converges to prospective Bayes risk, but they need more samples because they use more generic models of the stochastic process. This scenario is closely related to (online) multitask/meta-learning [12].

**Scenario 3 (Data is neither independent nor identically distributed).** Formally, this scenario consists of general stochastic processes. As an example, consider a Markov process $\mathbb{P}(Y_{t+1} = k \mid Y_t = k) = \theta$ with two states $k \in \{0, 1\}$ and $Y_1 \sim \text{Bernoulli}(\theta)$. The invariant distribution of this Markov process is $\mathbb{P}(0) = \mathbb{P}(1) = 1/2$. Prospective Bayes risk is also equal to 1/2. For stochastic processes that have a invariant distribution, it is impossible to predict the next state infinitely far into the future and therefore it is impossible to prospect. The prospective Bayes risk is trivially chance levels. In such situations, the learner could consider losses that are discounted over time. For example, one could use a slightly different loss than the one in Eq. (1) to write

$$\bar{\ell}_t(h, Z) = (1 - \gamma) \sum_{s=t+1}^{\infty} \gamma^{s-t-1} \ell(h_s(X_s), Y_s) \tag{4}$$

for some $\gamma \in [0, 1)$. In this example, we can calculate the prospective Bayes risk analytically; see Appendix B.2. For $\gamma = 0.9$, $\theta = 0.1$ and the zero-one loss, limiting prospective Bayes risk is 0.357. Now consider a learner which computes the MLE of the transition matrix $\Gamma_t^{t'-t}$. It calculates $\mathbb{P}(Y_{t'} \mid y_t) = \hat{p}_{t'}$ where $[1 - \hat{p}_{t'}, \hat{p}_{t'}] = \Gamma_t^{t'-t}[1 - y_t, y_t]^\top$ and uses the hypothesis $h_{t'}(X_{t'}) = \mathbf{1}(\hat{p}_{t'} > 0.5)$ (ties broken randomly). We can see in Fig. 1 that this learner converges to the prospective Bayes risk. This example shows that if we model the changes in the data, then we can perform prospective learning. This scenario is closely related to certain continual learning problems [13, 14].

**Scenario 4 (Future depends upon the current prediction).** Problems where predictions of the learner affect future data are an interesting special case of Scenario 3. Prospective learning can also be used to address such scenarios. For $\theta_0, \theta_1 \in [0, 1]$, consider a Markov decision process (MDP) $\mathbb{P}(Y_{t+1} = j \mid Y_t = j', h_{t+1}(1) = k) = \theta_k$ if $j = j'$ and $1 - \theta_k$ otherwise. I.e., the prediction $h_{t+1}(X_{t+1}) = k$ (recall that $X_t = 1$ for all times) is the decision and the MDP remains in the same state with probability $\theta_k$. Prospective Bayes risk for this example is the same as that of the example in Scenario 3. We can construct a prospective learner using a variant of Q-learning to first estimate the hypothesis and then estimate the probability $\mathbb{P}(Y_{t'} \mid y_t)$ like Scenario 3 above to predict on future data at time $t'$. See Appendix B.3. Prospective risk of this learner converges to Bayes risk in Fig. 1. This scenario is closely related to reinforcement learning [15].

## 3 How is prospective learning related to other learning paradigms? [6]

**Distribution shift.** Prospective learning [16] is equivalent to PAC learning [17] in Scenario 1 when data is IID. Situations when this assumption may not be valid are often modeled as a distribution shift between train and test data [2]. Techniques such as propensity scoring [18, 19] or domain adaptation [20, 21] reweigh or map the train/test data to get back to the IID setting; techniques like domain invariance [22, 1] build a statistic that is invariant to the shift. Typically, the loss is unchanged across train and test data. If the set of marginals $\{\mathbb{P}(Z_t)\}$ of the stochastic process only has two elements, then PL is equivalent to the classical distribution shift setting. But otherwise, in PL, data is correlated across time, distributions (marginals) can shift multiple times, and risk changes with time.

**Multi-task, meta-, continual, and lifelong learning.** A changing data distribution could be modeled as a sequence of tasks. Depending upon the stochastic process, different concepts are relevant, e.g., multi-task learning [23] is useful for Scenario 2 and Appendix D when there are a finite number of tasks. Much of continual or lifelong learning [14, 13] focuses on "task-incremental" and "class-incremental" settings [24], in which the learner knows when the task switches. PL does not make this assumption, and therefore, the problem is substantially more difficult. "Data-incremental" (or task-agnostic) setting [25], is similar to PL. But the main difference is the goal: continual or lifelong learning seeks to minimize past error. As a consequence, continual learning methods are poor prospective learners; see Section 6. Online meta-learning [26–29] is close to task-agnostic continual learning, except that the former models tasks as being sampled IID from some distribution of tasks. Due to this, one cannot predict which task is next, and therefore cannot prospect.

**Sequential decision making and online learning.** PL builds upon works on learning from streaming data. But our goals are different. For example, Gama et al. [30] minimize the error on samples from a stationary process; Hayes et al. [31] minimize the error on a fixed held-out dataset or on all past data—neither of these emphasizes prospection. There is a rich body of work on sequential decision making, e.g., predicting a finite-state, stationary ergodic process from past data [32]. Even in this simple case, there does not exist a consistent estimator using the finite past $Z_{1:t}$ [33–35]. This is also true for regression [36, 37], when the true hypothesis $f^*$ s.t. $Y_t = f^*(X_t)$ is fixed. In other words, Bayes risk $R_t^*$ in Theorem 1 may be non-zero in PL even for finite-state, stationary ergodic processes. Hanneke [38] lifts the restriction on stationarity and ergodicity. They obtain conditions on the input process $X$ for consistent inductive (predict at time $t' > t$ using data up to $t$), self-adaptive (predict at time $t'$ using $Z_{\leq t}$ and $X_{t+1:t'}$) and online learning [39, 40] (predict at $t'$ using $Z_{\leq t'}$). They prove the existence of a learning rule that is consistent for every $X$ that admits self-adaptive learning. If $X$ is "smooth", i.e., input marginals have a similar support over time, then ERM has a similar sample complexity as that of the IID setting [41]. Haghtalab et al. [42] give algorithmic guarantees for several online estimation problems in this setting.

The true hypothesis in PL can change over time. This is different from the continual learning setting where we can find a common hypothesis for tasks at all time [43], and this is why our proofs work quite differently from existing ones in the literature. Instead of a hypothesis class $\mathcal{H} \subseteq \mathcal{Y}^{\mathcal{X}}$, we define the notion of a hypothesis class that consists of sequences of predictors, i.e., subset of $(\mathcal{Y}^{\mathcal{X}})^{\mathbb{N}}$; we can do ERM in this new space. Instead of consistency of prediction as in Hanneke [38], we give guarantees for strong learnability, i.e., convergence of the ERM risk to the Bayes risk.

**Information theory.** There are also works that have sought to characterize classes of stochastic processes that can be predicted fruitfully. Bialek et al. [44] defined a notion called predictive information (closely related to the information bottleneck principle [45]) and showed how it is related to the degrees of freedom of the stochastic process. Shalizi and Crutchfield [46] showed that a causal-state representation called an $\epsilon$-machine is the minimal sufficient statistic for prediction.

## 4 Theoretical foundations of prospective learning

**Definition 1** (**Strong Prospective Learnability**). A family of stochastic processes is strongly prospectively learnable, if there exists a learner with the following property: there exists a time $t'(\epsilon, \delta)$ such that for any $\epsilon, \delta > 0$ and for any stochastic process $Z$ from this family, the learner outputs a hypothesis $h$ such that $\mathbb{P}\left[R_t(h) - R_t^* < \epsilon\right] \geq 1 - \delta$, for any $t > t'$.

This definition is similar to the definition of strong learnability in PAC learning with one key difference. Prospective Bayes risk $R_t^*$ depends upon the realization of the stochastic process $z_{\leq t}$ up to time $t$. In PAC learning, it would only depend upon the true distribution of the data. Not all

---

[6]Also see Appendix A for a more elaborate discussion.

families of stochastic processes are strongly prospectively learnable. We therefore also define weak learnability with respect to a "chance" learner that predicts $\mathbb{E}[Y]$ and achieves a prospective risk $R_t^0$.[7]

**Definition 2** (**Weak Prospective Learnability**). A family of stochastic processes is weakly prospectively learnable, if there exists a learner with the following property: there exists an $\epsilon > 0$ such that for any $\delta > 0$, there exists a time $t'(\epsilon, \delta)$ such that for any stochastic process $Z$ from this family, $\mathbb{P}\left[R_t^0 - R_t(h) > \epsilon\right] \geq 1 - \delta$, for any $t > t'$.

In PAC learning for binary classification, strong and weak learnability are equivalent [47] in the distribution agnostic setting, i.e., when strong and weak learnability is defined as the ability of a learner to learn any data distribution. But even in PAC learning, if there are restrictions on the data distribution, strong and weak learnability are not equivalent [48]. This motivates Proposition 1 below. Before that, we define a time-agnostic empirical risk minimization (ERM)-based learner. In PAC learning, ERM selects a hypothesis that minimizes the empirical loss on the training data. It outputs a time-agnostic hypothesis, i.e., using data, say, $z_{\leq t}$ standard ERM returns the same predictor for future data from any time $t' > t$. There is a natural application of ERM to prospective learning problems, defined below.

**Definition 3** (**Time-agnostic ERM**). Let $\mathcal{H}$ be a hypothesis class that consists of time-agnostic predictors, i.e., $h_t = h_{t'}$ for any $t, t' \in \mathbb{N}$ for all predictors $h \in \mathcal{H}$. Given data $z_{\leq t}$, a learner that returns

$$\hat{h} = \arg\min_{h \in \mathcal{H}} \frac{1}{t} \sum_{s=1}^{t} \ell(s, h_s(x_s), y_s) \tag{5}$$

is called a time-agnostic empirical risk minimization (ERM)-based learner.

Time-agnostic ERM in prospective learning may use a time-dependent loss $\ell(s, h_s(x_s), y_s)$ upon the training data. This ERM is not very different from standard ERM in PAC learning (when instantiated with the hypothesis class that consists of sequences of predictors, that we are interested here). If data is IID (Scenario 1), then there is no information provided by time in the training samples. But if there are temporal patterns in the data, take Scenarios 2 and 3 or Scenario 4 as examples, then time-agnostic ERM as defined here will return predictors that are different than those of standard ERM that uses a time-invariant loss.

**Proposition 1.** *There exist stochastic processes for which time-agnostic ERM is not a weak prospective learner. There also exist stochastic processes for which time-agnostic ERM is a weak prospective learner but not a strong one.*

See Appendix E for the proof. We do not know yet whether (or when) strong and weak learnability are equivalent for prospective learning.

## 5 Prospective Empirical Risk Minimization (ERM)

In PAC learning, the hypothesis returned by ERM using the training data can predict arbitrarily well (approximate the Bayes risk arbitrarily well with arbitrarily high probability), with a sufficiently large sample size. This statement holds if (a) there exists a hypothesis in the hypothesis class whose risk matches the Bayes risk asymptotically, and (b) if risk on training data converges to that on the test data sufficiently quickly and uniformly over the hypothesis class [49, 50]. Theorem 1 is an analogous result for prospective learning.

**Theorem 1** (**Prospective ERM is a strong prospective learner**). *Consider a finite family of stochastic processes $\mathcal{Z}$. If we have (a) consistency, i.e., there exists an increasing sequence of hypothesis classes $\mathcal{H}_1 \subseteq \mathcal{H}_2 \subseteq \ldots$ with each $\mathcal{H}_t \subseteq (\mathcal{Y}^{\mathcal{X}})^{\mathbb{N}}$ such that $\forall Z \in \mathcal{Z}$,*

$$\lim_{t \to \infty} \mathbb{E}\left[\inf_{h \in \mathcal{H}_t} R_t(h) - R_t^*\right] = 0, \tag{6}$$

*where $h \in \mathcal{H}_t$ is a random variable in $\sigma(Z_{\leq t})$, and (b) uniform concentration of the limsup, i.e., $\forall Z \in \mathcal{Z}$,*

$$\mathbb{E}\left[\max_{h \in \mathcal{H}_t}\left|\bar{\ell}_t(h, Z) - \max_{u_t \leq m \leq t} \frac{1}{m} \sum_{s=1}^{m} \ell(s, h_s(x_s), y_s)\right|\right] \leq \gamma_t, \tag{7}$$

---

[7]We can also define weak learnability with respect to the prospective risk of a particular learner, even one that is not prospective. This may be useful to characterize learning for stochastic processes which do not admit strong learnability.

*for some $\gamma_t \to 0$ and $u_t \to \infty$ with $u_t \leq t$ (all uniform over the family of stochastic processes), then there exists a sequence $i_t$ that depends only on $\gamma_t$ such that a learner that returns*

$$\hat{h} = \arg\min_{h \in \mathcal{H}_{i_t}} \max_{u_{i_t} \leq m \leq t} \frac{1}{m} \sum_{s=1}^{m} \ell(s, h_s(x_s), y_s), \tag{8}$$

*is a strong prospective learner for this family. We define prospective ERM as the learner that implements Eq. (8) given train data $z_{\leq t}$.*

Appendix E.2 provides a proof, it builds upon the work of Hanneke [38]. The first condition, Eq. (6), is analogous to the consistency condition in PAC learning. In simpler words, it states that the Bayes risk can be approximated well using the chosen sequence of hypothesis classes $\{\mathcal{H}_t\}_{t=1}^{\infty}$. The second condition, Eq. (7), is analogous to concentration of measure in PAC learning, it requires that the limsup in Eq. (1) is close to an empirical estimate of the limsup (the second term inside the absolute value in Eq. (7)). At each time $t$, prospective ERM in Eq. (8) selects the best hypothesis $\hat{h} \in \mathcal{H}_t$[8] for future times $t' > t$, that minimizes an empirical estimate of the limsup using the training data $z_{\leq t}$. Prospective ERM can exploit the difference between the latest datum in the training set with time $t$ and the time for which it makes predictions $t'$ by selecting specific sequences inside the hypothesis class $\mathcal{H}_t$. For example, in Scenario 2 it can select sequences where alternating elements can be used to predict on data from even and odd times.

**Remark 1** (**How to implement prospective ERM?**). An implementation of prospective ERM is therefore not much different than an implementation of standard ERM, except that there are two inputs: time $s$ and the datum $x_s$. Suppose we use a hypothesis class where each predictor is a neural network, this could be a multi-layer perceptron or a convolutional neural network. The training set $z_{\leq t}$ consists of inputs $x_s$ along with corresponding time instants $s$ and outputs $y_s$. To implement prospective ERM, we modify the network to take $(s, x_s)$ as input (using any encoding of time, we discuss one in Section 6) and train the network to predict the label $y_s$. In Eq. (8) we can set $u_{i_t} \equiv t$, doing so only changes the sample complexity. At inference time, this network is given the input $(t', x_{t'})$ to obtain the prediction $y_{t'}$. Note that if prospective ERM is implemented in this fashion, the learner need not explicitly calculate the infinite sequence of predictors.[9]

**Corollary 1.** *There exist stochastic processes for which time-agnostic ERM is not a strong prospective learner, but prospective ERM is a strong learner.*

**Remark 2** (**Why we need an increasing sequence of hypothesis classes** $\mathcal{H}_1 \subseteq \mathcal{H}_2 \ldots$). We could have chosen $\mathcal{H}_t = \mathcal{H}_{t'}$ for all $t, t' \in \mathbb{N}$ to set up Theorem 1. But since the learner does not have a lot of data at early times, it should use a small hypothesis class. Just like PAC learning, the sequence $(\gamma_t)_{t \in \mathbb{N}}$ in Eq. (7) determines the convergence rate of a prospective learner. Therefore, using a monotonically increasing sequence of hypothesis classes is useful to ensure a good sample complexity.

**Theorem 2.** *Consider a finite family of stochastic processes $\mathcal{Z}$. If there exists a countable hypothesis class $\mathcal{H}$ such that*

$$\lim_{t \to \infty} \mathbb{E}\left[\inf_{h \in \mathcal{H}} R_t(h) - R_t^*\right] = 0, \tag{9}$$

*for any stochastic process $Z \in \mathcal{Z}$, where $h \in \mathcal{H}$ is a random variable in $\sigma(Z_{\leq t})$, then there exist $\mathcal{H}_t$, $u_t$, and $\gamma_t$ such that the two conditions of Theorem 1 are satisfied for this family.*

Appendix E.3 provides a proof. This theorem provides a concrete example for which the assumptions of Theorem 1 are satisfied. In PAC learning, one first proves uniform convergence for a finite hypothesis class. This can then be used to, say, calculate the sample complexity of ERM, or extended to infinite hypothesis classes using constructions such as VC-dimension and covering numbers [51]. The above theorem should be understood in the same spirit. It is a step towards characterizing the sample complexity of prospective learning.

Appendix C proves an analogue of Theorem 1 for prospective learning problems with discounted losses. Appendix D provides illustrative examples of prospective ERM for periodic processes and hidden Markov processes. For periodic processes, we can also calculate the sample complexity.

---

[8]Note that this hypothesis class has infinite sequences, $\mathcal{H}_t \subseteq \left(\mathcal{Y}^{\mathcal{X}}\right)^{\mathbb{N}}$.

[9]Hereafter, when we write ERM in empirical studies, we will mean a learner that approximates ERM via stochastic gradient descent.

# 6 Experimental Validation

This section demonstrates that we can implement prospective ERM on prospective learning problems constructed on synthetic data, MNIST and CIFAR-10. In practice, prospective ERM may approximately achieve the guarantees of Theorem 1. We will focus on the distribution changing, independently or not (Scenarios 2 and 3). Recall that Scenario 1 is the same as the IID setting used in standard supervised learning problems. Scenario 4 is more involved (see an example in Appendix B.3) and, therefore, we leave more elaborate experiments for future work. We discuss experiments that check whether large language models can do prospective learning in Appendix H.

**Learners and hypothesis classes.** Task-agnostic online continual learning methods are the closest algorithms in the literature that can address situations when data evolves over time. We use the following three methods.

(i) **Follow-the-Leader** minimizes the empirical risk calculated on all past data and is a no-regret algorithm [52]. We note that while this is a popular online learning algorithm, we do not implement the algorithm in an online fashion.

(ii) **Online SGD** fine-tunes the network using new data in an online fashion. At every time-step, weights of the network are updated once using the last eight samples.

(iii) **Bayesian gradient descent** [53] is an online continual learning algorithm designed to address situations where the identity of the task is not known during both training and testing, i.e., it implements continual learning without knowledge of task boundaries.

These three methods are not explicitly designed for prospective learning but they are designed to address the changing data distribution $t$.[10] We calculate the prospective risk of the predictor returned by these methods; note that they do not output a time-varying predictor and consequently, these methods output a time-agnostic hypothesis. As a result, when we evaluate the prospective risk of these methods, we use the same hypothesis for all future time. For all three methods, we use a multi-layer perceptron (MLP) for synthetic data and MNIST, and a convolutional neural network (CNN) for CIFAR-10.

For **prospective ERM** the sequence of predictors is built by incorporating time as an additional input to an MLP or CNN as follows. For frequencies $\omega_i = \pi/i$ for $i = 1, \ldots, d/2$, we obtain a $d$-dimensional embedding of time $t$ as $\varphi(t) = (\sin(\omega_1 t), \ldots, \sin(\omega_{d/2} t), \cos(\omega_1 t), \ldots, \cos(\omega_{d/2} t))$. This is similar to the position encoding in Vaswani et al. [55]. A predictor $h_t(\cdot)$ uses a neural network that takes as input, an embedding of time $\varphi(s)$, and the input $x_s$ to predict the output $y_s$ for any time $s \in \mathbb{N}$. Using such an embedding of time is useful in prospective learning because, then, one need not explicitly maintain the infinite sequence of predictors $h \equiv (h_1, \ldots, )$.

**Training setup.** We use the zero-one error $\mathbf{1}\{\hat{y} \neq y\}$ to calculate prospective risk for all problems; all learners are trained using a standard surrogate of this objective, the cross-entropy loss. For all experiments, for each time $t$, we calculate the prospective risk $R_t(h)$ in Eq. (2) of the hypothesis created by these learners for a particular realization of the stochastic process $z_{\leq t}$. For each prospective learning problem, we generate a sequence of 50,000 samples. Learners are trained on data from the first $t$ time steps ($z_{\leq t}$) and prospective risk is computed using samples from the remaining time steps. Except for online SGD and Bayesian gradient descent, learners corresponding to different times are trained completely independently. See Appendix F for more details.

**Remark 3** (**Why we do not use existing benchmark continual learning scenarios**). The tasks constructed below resemble continual learning benchmark scenarios such as Split-MNIST or Split-CIFAR10 [56] where data from different distributions are shown sequentially to the learner. However, there are three major differences. First, in these existing benchmark scenarios, data distributions do not evolve in a predictable fashion, and prospective learning would not be meaningful. Second, existing scenarios consider a fixed time horizon. We are keen on calculating the prospective risk for much longer horizons whereby the differences between different learners are easier to discern; our experiments go for as large as 30,000 time steps. Third, our tasks have the property that the Bayes optimal predictor changes over time.

---

[10]There are many algorithms in the existing literature that the reader may think of as reasonable baselines. We have chosen a representative and relevant set here, rather than an exhaustive one. For example, online meta-learning approaches are close to online-SGD; since the learner fine-tunes on the most recent data. Algorithms in the literature on time-series (i) focus on predicting future data, say, $Y_{t'}$ given past data $y_{\leq t}$ without taking covariates $X_{t'}$ or some exogenous variables $X_{\leq t}$ into account, (ii) can usually only make predictions for a pre-specified future context window [54], and (iii) work for low-dimensional signals (unlike images).

## 6.1 Prospective learners for independent but not identically distributed data (Scenario 2)

We create tasks using synthetic data, MNIST and CIFAR-10 datasets to design prospective learning problems when data are independent but not identically distributed across time (Scenario 2).

**Dataset and Tasks.** For the synthetic data, we consider two binary classification problems ("tasks") where the input is one-dimensional. Inputs for both tasks are drawn from a uniform distribution on the set $[-2, -1] \cup [1, 2]$. Ground-truth labels correspond to the sign of the input for Task 1, and the negative of the sign of the input for Task 2. For MNIST and CIFAR-10 we consider 4 tasks corresponding to data from classes 1-5, 4-7, 6-9 and 8-10 in the original dataset, i.e., the first task considers classes 1-5 labelled 1-5 respectively, the second task considers classes 4-7 labelled 1-4, the third task considers classes 6-9 labeled 1-4 and the last task considers labels 8-10 labelled 1-3. In other words, images from class 1 in task 1, class 4 from task 2 and class 6 from task 3 are all assigned the label 1. For the prospective learning problem based on synthetic data, the task switches every 20 time steps. For MNIST and CIFAR-10, the data distribution cycles through the 4 tasks, and the distribution of data changes every 10 time-steps. For more details, see Appendix F.

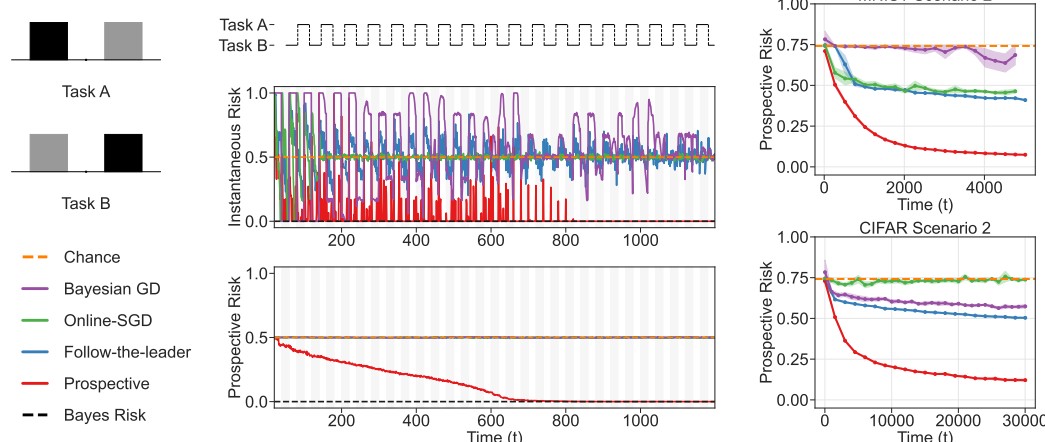

**Figure 2: Prospective ERM can achieve good instantaneous and prospective risk in Scenario 2. Left:** Instantaneous and prospective risks for problems constructed using synthetic data (see text) across 5 random seeds (which govern the sequence of samples and the weight initializations of neural networks). Instantaneous risk spikes when the task switches for many online learning baseline algorithms. In contrast, prospective ERM has minimal spikes at later times and both instantaneous and prospective risks eventually converge to zero. **Right:** Prospective risk for different baseline algorithms and prospective ERM for tasks constructed using MNIST and CIFAR-10 for Scenario 2. In all three cases, the risk of prospective ERM approaches Bayes risk while online learning baselines considered here do not achieve a low prospective risk. For comparison, the chance prospective risk is 0.5 for synthetic data and 0.742 for MNIST and CIFAR-10 tasks.

Fig. 2 shows that **prospective ERM can learn problems when data are independent but not identically distributed (Scenario 2)**. For prospective learning problems constructed from synthetic data, the risk of prospective ERM converges to prospective Bayes risk over time. For the MNIST and CIFAR-10 prospective problems, the prospective learning risk drops precipitously. In contrast, online learning baselines discussed above achieve a far worse prospective risk. Observe that Follow-the-Leader (blue) performs as well, or better, as online SGD and Bayesian GD. This is not surprising, while ERM models corresponding to each time $t$ were trained independently, networks corresponding to online SGD and Bayesian GD were training in an online fashion; in practice it is often difficult to tune online learning methods effectively [57].[11]

## 6.2 Prospective learners when data are neither independent nor identically distributed (Scenario 3)

**Dataset and Tasks.** For synthetic data, we construct 4 binary classification problems with two-dimensional input data (see Fig. 3 and caption for details). For CIFAR-10 and MNIST, we consider four tasks corresponding to the classes 1-5, 4-7, 6-9 and 8-10. Using these tasks, we construct

---

[11]For CNNs on CIFAR-10, if one concatenates the time embedding directly to the input images as opposed to concatenating to a layer before softmax, like it is done here, the prospective risk in Fig. 2 (right) is much higher (worse by almost 0.2. See Fig. A.6). The implementation details of time embedding do matter when implementing prospective learners in practice, even if Theorem 1 is true in general.

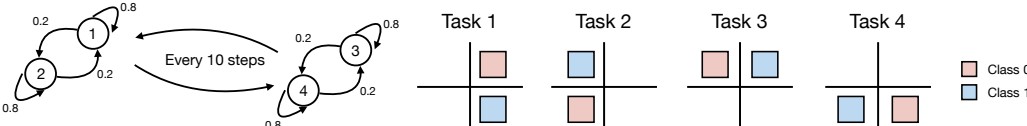

**Figure 3: Left:** For MNIST and CIFAR-10, we consider 4 tasks corresponding to the classes 1-5, 4-7, 6-9 and 8-10. Using these tasks, we construct Scenario 3 problems corresponding to a stochastic process which is a hierarchical hidden Markov model. After every 10 time-steps, a different Markov chain governs transitions among tasks (one Markov chain for tasks 1 and 2, and another for tasks 3 and 4). This ensures that the stochastic process does not have a stationary distribution. **Right:** For synthetic data, the 4 tasks are created using two-dimensional input data as shown pictorially above. The four parts of the input domain are $\{(x_1, x_2) : 1 \leq x_1, x_2 \leq 2\}$, $\{(x_1, x_2) : 1 \leq x_1 \leq 2, \text{ and } -2 \leq x_2 \leq -1\}$, $\{(x_1, x_2) : -2 \leq x_1, x_2 \leq -1\}$ and $\{(x_1, x_2) : -2 \leq x_1 \leq -1 \text{ and } 1 \leq x_2 \leq 2\}$. Colors indicate classes. The hierarchical hidden Markov model for transitions among the tasks is identical to the MNSIT and CIFAR-10 setting shown on the left.

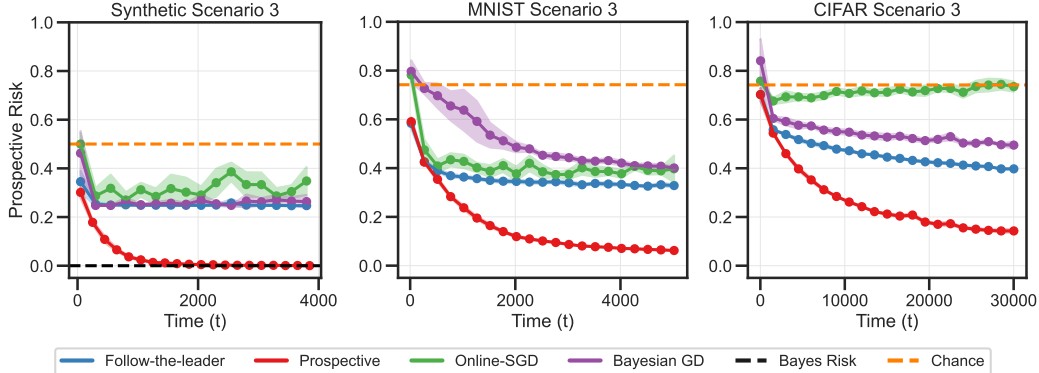

**Figure 4: Prospective ERM can achieve good prospective risk in Scenario 3.** Prospective risk across 5 random seeds (which govern the sequence of samples and the weight initializations of neural networks). In all three cases, the risk of prospective ERM approaches Bayes risk while a number of baseline algorithms do not achieve a low prospective risk. Stochastic processes in these problems corresponding to Scenario 3 do not have an invariant distribution. This is why a time-agnostic hypothesis (ERM) that is constructed by the baseline algorithms does not achieve a good prospective risk.

problems where the data distribution is governed by a stochastic process which is a hierarchical hidden Markov model (Scenario 3). After every 10 time-steps, a different Markov chain governs transitions among tasks (one Markov chain for tasks 1 and 2, and another for tasks 3 and 4, as shown in Fig. 3). These choices ensure that the stochastic process does not have a stationary distribution.[12]

As Fig. 4 shows, **prospective ERM can prospectively learn problems when data is both independent and not identically distributed (Scenario 3**. Stochastic processes in these problems corresponding to Scenario 3 do not have a stationary distribution. This is why a time-agnostic hypothesis (Follow-the-Leader) does not achieve a good prospective risk, unlike prospective ERM. Appendix G discusses additional experiments for Scenario 3 for different kinds of Markov chains.

## 7  Discussion

Prospective learning, as we see it, is a paradigm of learning that characterizes many real-world scenarios which are currently modeled using much stronger (and less accurate) assumptions. These simplifying assumptions have certainly enabled progress in machine learning. But systems deployed built upon these approaches have proven to be extremely fragile in certain real-world settings. Today's machine learning-based systems fail to track changes in the data. They certainly do not model how biological organisms learn robustly and effectively over time. We believe characterizing which kinds of stochastic processes are prospectively learnable under which kinds of time-sensitive loss functions will be an important next theoretical step. Developing algorithms, from the perspective of prospective learning, which have theoretical guarantees in practice, will be another next step. Finally, building algorithms that scale and can therefore be deployed in real-world systems, will be important to demonstrate the utility of this approach. The precise real-world applications in which prospective learning based methods will outperform PAC learning, remains to be seen.

---

[12]As we discussed in Scenario 3, prospective Bayes risk can be trivial in situations when the stochastic process has a stationary distribution.

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

## Acknowledgement

This work was funded by grants from the National Science Foundation (IIS-2145164, CCF-2212519) and the Office of Naval Research (ONR) N00014-22-1-2255. RR was funded by a fellowship from AWS AI to Penn Engineering's ASSET Center for Trustworthy AI. ADS was supported by a fellowship from the Mathematical Institute for Data Science (MINDS) at JHU. We are grateful to all those who provided helpful feedback on earlier drafts of this work, including Marlos Machado.

## A  Isn't this just...

When we describe prospective learning to people the first time, they often wonder how it is different—both conceptually and formally—from other previously established learning frameworks. In fact, for many of them, the English language descriptions are seemingly identical to those which describe prospective learning. However, the English language is often imprecise and this has created a lot of confusion among both theoreticians and practitioners as to the precise differences, potential benefits and pitfalls, between different learning frameworks. Here, we provide detailed formal distinctions between prospective learning and other related learning frameworks. Table A.1 summarizes the key distinctions between several machine learning frameworks, with further details provided below. The key difference between prospective learning and all other learning frameworks mentioned below is that in prospective learning, the hypothesis can make an inference (or take an action) arbitrarily far in the future. Certain versions of forecasting also have that property (probabilistic forecasting [58]), but forecasting has several other distinctions.

**Table A.1: Comparison of different machine learning frameworks** in terms of the distributional assumptions on the data. Task IID indicates that data within a task are IID, and that tasks are IID from some meta-distribution. Loss characterizes whether the assumed loss function is instantaneous or time-varying. Optimal hypothesis indicates the total possible number of different optimal hypotheses (assuming each hypothesis has a unique risk). Data availability indicates whether the data are available all at once (in batch), or after each task arrives (task sequential), or one data sample at a time (sequential). The answers are given for typical settings, further details are available in the paragraphs below.

| Framework | Data Distribution | Loss | # Optimal hypotheses | Data Availability |
|---|---|---|---|---|
| PAC Learning | IID | Instantaneous | 1 | Batch |
| Transfer Learning | Change Point | Instantaneous | 1-2 | Batch |
| Meta Learning | Task IID | Instantaneous | # Tasks | Batch/Sequential |
| Lifelong Learning | Task IID | Instantaneous | # Tasks | Task Sequential |
| Online Learning | None | Instantaneous | 1 | Sequential |
| Forecasting | Stochastic Process | Fixed horizon | # Time steps | Batch/Sequential |
| Reinforcement Learning | Markov Decision Process | Time-varying | 1 | Sequential |
| Prospective Learning | Stochastic (Decision) Process | Instant./Time-varying | # Time steps | Any |

### A.1  ...PAC learning?

PAC learning [59] is a special case of prospective learning when the stochastic process is time-invariant (meaning the data are IID) and the loss is fixed. Also, it is only concerned with batch data. It is an interesting question as to whether prospective learning as we have defined here is useful for IID data. We do not know yet in general. In Appendix B.1, we provide a simple example where prospection turns out to be beneficial, even in the IID setting. More broadly, we wonder whether the viewpoint proposed in this paper might lead to novel algorithms for solving learning problems on IID data that do not have closed form solutions.

### A.2  ...transfer learning?

In transfer learning [21], including domain (covariate) shift (adaptation) [20, 60], and out-of-distribution (OOD) [61] learning, there are two distributions, a source and a target distribution; thus, the distribution changes only once, rather than potentially once per time step. Depending on whether the goal is to perform well only on the target, or both the source and the target, there are one or two optimal hypotheses. Also, that the distribution has changed is often known (though not always, as in OOD learning).

### A.3  . . . meta-learning?

Meta-learning [62, 63] is similar to multi-task learning [64], and includes as special cases zero-shot [65] and few-shot learning [66]. Here, the data are *Task IID*, meaning that the distribution within a task is IID, and the distribution of tasks themselves is also IID, rendering it impossible to predict future distributions very well (the best one could do is guess the next task is whichever task is most likely). Typically, that the task/distribution changes is known, but not always. In classical meta-learning, data are available in one batch, but in online meta-learning, data are sequentially available [67]. The goal is to perform well on the next (unknown) distribution, as opposed to all future (unknown) distributions as in prospective learning.

### A.4  . . . lifelong learning?

Lifelong (continual) learning [14, 13, 68] is nearly identical to online meta-learning [67]. However, the data are typically available in one batch per task. The goal is also a bit different, rather than performing well on the next distribution, in lifelong learning, the goal is also to continue performing well on previous distributions. Often, the learner is aware that the distribution shifted [24], but not always [25].

### A.5  . . . online learning?

A key property of online learning [69] is that there are no distributional assumptions [70], and therefore, the goal is not about generalization error. Instead, performance is evaluated relative to the best a fixed hypothesis could have done up until now. Also, in online learning, the environment is often adversarial.

### A.6  . . . forecasting?

Forecasting [71], time-series or sequential analysis [72, 73] assume the data follow a stochastic process, much like prospective learning often does (e.g., Scenarios 2 and 3), and therefore, the number of optimal hypothesis can be equal to the number of time steps. However, in forecasting, the loss is associated with a fixed (pre-specified) horizon, or several horizons [54]. Forecasting also often assumes a parametric model, but not always [74, 75]. Probabilistic forecasting [58] can also predict arbitrarily far in the future by iteratively updating its probabilistic forecasts. However, this is prone to numerical errors, as evidenced in sequential Monte Carlo.

### A.7  . . . reinforcement learning?

Reinforcement learning (RL) [15] is only concerned with situations where there is a control element, that is, the hypothesis chooses an action (which potentially impacts future distributions), rather than merely an inference (which does not). Thus, it excludes Scenarios 2 and 3. Moreover, RL theory focuses on Markov Decision Processes [76], whereas PL operates on larger classes of stochastic decision processes, including non-Markov processes (e.g., see examples of non-Markov stochastic processes in Scenario 3). Depending on context, PL also considers loss functions that are instantaneous. Classical RL assumes data are sequentially available, yet offline RL operates in batch mode [77]. Perhaps most importantly, in classical RL, the optimal hypothesis (policy) is not time-varying, though recent generalizations are forthcoming [78]. Also, in RL, there are typically many episodes, whereas in prospective learning there is only a single episode (though single-episode RL is also forthcoming [79]).

To elaborate on the first point above, assume that our decisions do not impact the future, but the optimal hypothesis is time-dependent, that is $h_t^* \neq h_{t'}^*$ for some $t \neq t'$. Why would we care about the risk for any $t' > t$ (like RL, but not like online/continual learning), given that our decision at time $t$ does not impact $Z_{t'}$ at all? We only ever incur the current loss, that is, $\ell(t, h_t(x_t), y_t)$. So, it would seem that as long as we minimize this current loss, there is no reason to care about any future loss. First note that minimizing the expected cumulative future loss is sufficient for minimizing the loss averaged over a finite future horizon, this is formally shown in Corollary 2. But more importantly, these two problem settings are rather different. Minimizing the prospective risk (expected cumulative future loss) forces the learner to learn/model all the different modes of variation in the data. Missing even a small (low energy) mode of variation can lead to large prospective risk. If the learner only seeks to minimize the current loss, it need not have any representation of how data evolves over time. It will not be a good prospective learner. Recall that online learners (which minimize the current loss) have large worst case regret. Prospective learning effectively evaluates the regret over the infinite future horizon. The two settings are closely related if data evolves slowly, as argued in Fakoor et al. [19].

## A.8 ... recurrent neural networks?

Recurrent neural networks (RNNs), including Long Short Term Memory (LSTM) networks [80], as well as echo state machines and liquid state machines (and other reservoir computing techniques [81]), and Gaussian Processes [82] seem like they are solving prospective learning problems. Indeed, they are all reasonable architectures for satisfying the conditions of Theorem 1. Insofar as they do satisfy those conditions, then they are indeed prospective learners.

# B  Calculations for scenarios in Section 2.1

## B.1  Can learning benefit from prospection in the IID scenario?

Consider Scenario 1 where the learner returns the hypothesis $h_{t'} = h_t = \text{threshold}(\hat{p}_t > 0.5)$ for all $t' > t$, where $\hat{p}_t = \frac{1}{t} \sum_{s=1}^{t} y_s$ is the maximum likelihood estimator (MLE). Alternatively, if we assume a prior distribution $\text{Beta}(\alpha, \beta)$ over $p$, then the resulting maximum *a posteriori* (MAP) estimate is $\hat{p}_t = \frac{\alpha + \sum_{s=1}^{t} z_s - 1}{\alpha + \beta + t - 2}$. We define a second prospective learner based on MAP that returns the sequence $\hat{h}_{\geq t} = (\hat{h}_t, \hat{h}_t, \cdots)$, where $\hat{h}_t = \text{threshold}(\hat{p}_t > 0.5)$ for all future times beyond $t$. If the prior distribution has a small divergence with respect to the true posterior distribution, then the second learner converges faster to the Bayes optimal risk; for a poor choice of prior, the convergence is slower. However, in such situations, we show that we can modify the MAP-based learner to use prospection and incorporate "time" to result in faster convergence to the Bayes risk.

Let $y_1, \ldots, y_t$ be the IID sample sequence observed up to time $t$. The idea here is to compute the rate of change $\Delta p(t)$ of the MAP estimate at time $t$ which is given by,

$$\Delta p(t) = \text{MAP}(y_1, \ldots, y_t) - \text{MAP}(y_1, \ldots, y_{t-1}) \tag{10}$$

Taking expectation on both sides of Equation (10), and plugging in $\hat{p}_t = \text{MAP}(y_1, \ldots, y_t)$ for the true parameter $p$, we construct the following estimate for $\Delta p(t)$.

$$\hat{\Delta} p(t) = \frac{(\alpha - 1) + tp}{(\alpha - 1) + (\beta - 1) + t} - \frac{(\alpha - 1) + (t - 1)p}{(\alpha - 1) + (\beta - 1) + t - 1}$$

Using this rate of change, we may forecast the estimate $p_{t'}$ at time $t' > t$ as follows.

$$\hat{p}_{t'} = \hat{p}_t + \sum_{s=t}^{t'-1} \hat{\Delta} p(s)$$

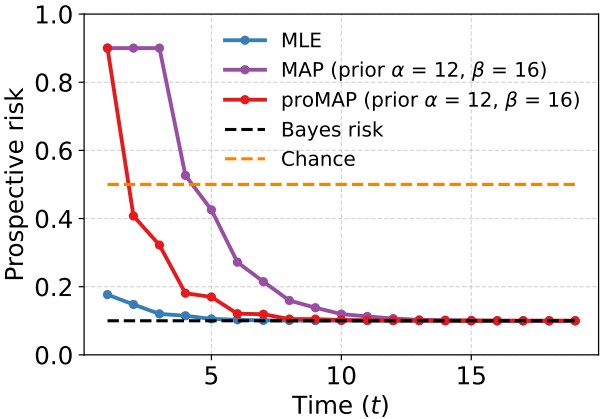

**Figure A.1:** Prospective risk of MLE *(blue)*, MAP *(purple)*, prospective MAP *(red)* and random-chance *(orange)* based learners with respect to time. Both MAP and prospective MAP estimators assume a prior distribution of $\text{Beta}(12, 16)$ over $p$.

We refer to this as the prospective MAP estimate. Based on it, we set the hypothesis to be $h_{t'} = \text{threshold}(\hat{p}_{t'} > 0.5)$ for all future times beyond $t$. In Figure A.1, we plot the prospective risk the MLE, MAP, and prospective MAP-based learners. Due to an unfavorable prior, the MAP-based learner converges slowly. However, prospective MAP-based learner manages to leverage its forecasting to achieve a faster convergence rate despite having the same prior as the MAP. This shows that we can indeed benefit from prospection even in the IID case.

## B.2 Bayes risk for a Markov chain

We would like to compute the prospective Bayes risk, when the evolution of the samples is governed by a Markov transition matrix where $P(Y_{t+1} = 0 \mid Y_t = 0) = \theta_0$ and $P(Y_{t+1} = 1 \mid Y_t = 1) = \theta_1$, i.e., the transition matrix is

$$\Gamma = \begin{bmatrix} \theta_0 & 1 - \theta_0 \\ 1 - \theta_1 & \theta_1 \end{bmatrix}.$$

The probability distribution at time $t'$ is given by $\Gamma^{t'-t}(z_t, 1 - z_t)^T$. The eigenvalues of the transition

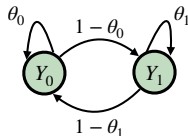

**Figure A.2:** Markov chain describing the evolution of data

matrix are $\lambda_1 = 1$ and $\lambda_2 = \theta_0 + \theta_1 - 1$ with the corresponding eigenvectors being $(1, 1)^\top$ and $(\theta_0 - 1, 1 - \theta_1)^\top$. Diagonalizing $\Gamma$ we get

$$\Gamma^{t'-t} = \begin{bmatrix} 1 & \theta_0 - 1 \\ 1 & 1 - \theta_1 \end{bmatrix} \begin{bmatrix} \lambda_1^{t'-t} & 0 \\ 0 & \lambda_2^{t'-t} \end{bmatrix} \begin{bmatrix} 1 & \theta_0 - 1 \\ 1 & 1 - \theta_1 \end{bmatrix}^{-1}$$

$$= \frac{1}{(2 - \theta_0 - \theta_1)} \begin{bmatrix} 1 - \theta_1 + (1 - \theta_0)\lambda_2^{t'-t} & (1 - \theta_0) - \lambda_2^n \\ 1 + \theta_1 + (1 - \theta_0)\lambda_2^{t'-t} & (1 - \theta_0) + \lambda_2^n. \end{bmatrix}$$

which implies that the probability distribution of the state at time $t'$ is

$$\pi_{t'} = \frac{1}{(2 - \theta_0 - \theta_1)} \begin{bmatrix} 1 - \theta_1 \\ 1 - \theta_0 \end{bmatrix} + \frac{\lambda_2^{t'-t}\left((1 - \theta_0)(1 - z_t) - (1 - \theta_1)z_t\right)}{(2 - \theta_0 - \theta_1)} \begin{bmatrix} 1 \\ -1 \end{bmatrix}.$$

Hence, the optimal sequence of hypotheses is $h^*_{\geq t+1} = (h^*_{t+1}, h^*_{t+2}, \dots)$, where

$$h^*_{t'} = \arg\max_{i \in \{0,1\}} \pi_{t'}(i)$$

with Bayes risk equal to $R^*_t = \lim_{T \to \infty} \frac{1}{T} \sum_{s=t}^{T} \min_{i \in \{0,1\}} \pi'_t(i)$. This reduces to

$$R^*_t = \frac{1}{(2 - \theta_0 - \theta_1)} \min(1 - \theta_0, 1 - \theta_1);$$

the second term in the expression of $\pi_{t'}$ vanishes as $T \to \infty$. If $\theta_0 = 0.9$ and $\theta_1 = 0.5$, then $R^*_t = 1/6$.

If we restrict our attention to the case where $\theta_0 = \theta_1$, the discounted Bayes risk reduces to

$$(1 - \gamma) \sum_{s=t+1}^{\infty} \gamma^{s-t-1} \ell(h^*_s) = (1 - \gamma) \sum_{s=t+1}^{\infty} \left( \frac{\gamma^{s-t-1}}{2} - \frac{\gamma^{s-t-1}\left|\lambda_2^{s-t}\right|}{2} \right) = (1 - \gamma) \left( \frac{1}{2(1 - \gamma)} - \frac{|\lambda_2|}{2(1 - |\lambda_2|\gamma)} \right)$$

Substituting $\theta_0 = \theta_1 = 0.1$, the discount risk for $\gamma = 0.9$ is $0.357$.

## B.3 Prospective learning in Scenario 4 when the future depends upon the current prediction

There are two types of prospective learners—one that passively observes the environment and makes inferences and another that acts on the environment and influences it. Scenario 1, Scenario 2, Scenario 3 fall into the first category which is the primary emphasis of our paper. Scenario 4 presents a prospective learning problem where the learner can influence the future realizations of the stochastic process through its decisions.

Our prospective learner is inspired from reinforcement learning, where the current state is $Y_{t-1}$, the action is $h_t$ and the next state is $Y_t$. The reward at the $t^{\text{th}}$ time-step is $r(t, h_{t+1}, y_{t+1}) = \mathbf{1}\{h_{t+1} = y_{t+1}\}$ as a result of selecting action $h_{t+1}$ given that the previous output was $y_t$, and next output $y_{t+1}$. The learner estimates the transition matrix corresponding to the MDP for each decision $h_{t+1}(X_{t+1}) \in \{0, 1\}$ using a similar procedure as that of Scenario 3,

$$\forall k \in \{0, 1\} : \hat{\Gamma}_t(k) = \begin{bmatrix} \hat{\theta}_k & 1 - \hat{\theta}_k \\ 1 - \hat{\theta}_k & \hat{\theta}_k \end{bmatrix}$$

where $\hat{\theta} \equiv \hat{\theta}_t \in [0, 1]^2$ after $t$ time steps. Using this estimate of $\hat{\Gamma}_t$, the learner solves for the value function (corresponding to the discounted prospective risk) that satisfies

$$\hat{Q}_t(y_t, h_{t+1}) = \sum_{y \in \{0,1\}} \underbrace{\mathbb{P}(Y_{t+1} = y \mid Y_t = y_t, h_{t+1} = h)}_{= \hat{\Gamma}_t(h_{t+1})_{y_t, y}} \left( r(t, h_{t+1}, y) + \gamma \max_{\bar{h}} \hat{Q}_t(y, \bar{h}) \right)$$

The value function can be solved using value iteration, iteratively until convergence. For a given $\hat{\Gamma}$, Banach's fixed point theorem guarantees this procedure will converge to the optimal value function in the tabular setting [83]. Once we have the Q-values $\hat{Q}_t$, we can use it to take actions. The optimal action at time $t'$ is $\arg\max_h Q(y_{t'}, h)$. However, unlike reinforcement learning, we do not know $y_{t'}$ for $t' > t$ and we must instead make a sequence of decisions using state $y_t$. The sequence of decision made by the learner is $\hat{h}_{>t} = (\hat{h}_{t+1}, \dots)$ where

$$\hat{\pi}_{t'}^\top = \pi_t^\top \prod_{s=t+1}^{t'} \hat{\Gamma}_t(\hat{h}_{s-1}),$$

$\pi_t = (1 - y_t, y_t)^\top$, i.e., $\pi_{t'}$ is the estimated distribution over the state at time $t'$, and

$$\hat{h}_{t'+1} = \arg\max_h \pi_{t'}^\top \hat{Q}_t(\cdot, h).$$

In Fig. 1, we have used $\theta_0 = \theta_1 = 0.1$ and a discount factor $\gamma = 0.9$. We find that this learner approaches the Bayes risk (0.357 which we calculate in Appendix B.2).

## C  Prospective ERM for discounted losses

Like we discussed in Scenario 3, in order to prospect meaningfully for some stochastic processes, we might need to consider a discounted future loss, e.g., the one in Eq. (4). Theorem 1 was proved only for the averaged future loss in Eq. (1). Here, we sketch out the proof of an analogous theorem for the discounted loss. Let

$$\ell_t^{(\tau)}(h, Z; \gamma) = \left( \frac{1 - \gamma}{1 - \gamma^{\tau+1}} \right) \sum_{s=t+1}^{t+\tau} \gamma^{s-t-1} \ell(h_s(x_s), y_s),$$

where $\ell : \mathcal{Y} \times \mathcal{Y} \mapsto [0, 1]$ is a bounded loss function and $0 < \gamma < 1$ is a constant. In general, we can use a probability measure $\mu^{(\tau)}$ supported on integers $\{1, \dots, \tau\}$ to write the loss as

$$\ell_t^{(\tau)}(h, Z; \mu^{(\tau)}) = \sum_{s=t+1}^{t+\tau} \mu_{s-t}^{(\tau)} \ell(h_s(x_s), y_s). \tag{11}$$

The averaged loss in Eq. (1) corresponds to $\mu_s^{(\tau)} = 1/\tau$ for all $s$. The discounted loss above corresponds to $\mu_s^{(\tau)} = \left( \frac{1-\gamma}{1-\gamma^{\tau+1}} \right) \gamma^{s-1}$.

In prospective learning, we are interested in the case when $\tau \to \infty$ and therefore let us define

$$\bar{\ell}_t(h, Z; \mu) = \limsup_{\tau \to \infty} \ell_t^{(\tau)}(h, Z; \mu^{(\tau)}).$$

where $\mu$ denotes the collection $\{\mu^{(\tau)}\}_{\tau=1}^\infty$. We can define $R_t(h; \mu)$ and $R_t^*(\mu)$ for this discounted loss using similar expressions as those in Eqs. (2) and (3). For clarity, let us use the notation $R_t(h, 1/\tau) \equiv R_t(h)$ and $R_t^*(1/\tau) \equiv R_t^*$ for the prospective risk and prospective Bayes risks corresponding to the averaged loss corresponding to $\mu_s^{(\tau)} = 1/\tau$.

**Corollary 2** (**Prospective ERM is a strong learner with discounted losses**). *Let the assumptions of Theorem 1 hold. If there exists a constant $c > 0$ such that $\forall Z \in \mathcal{Z}$,*

$$R_t(h; \mu) - R_t^*(\mu) \le c \left( R_t(h, 1/\tau) - R_t^*(1/\tau) \right) \quad \forall t \in \mathbb{N}, h \in \mathcal{H}_t \tag{12}$$

*i.e., if gap in the risk for the discounted loss is dominated uniformly by the gap in the risks for the averaged loss (over all realizations of the stochastic process), then prospective ERM implemented in Eq. (8) (implemented with the averaged loss) is a strong prospective learner, i.e., its discounted risk $R_t(\hat{h}, \mu)$ converges to the discounted Bayes risk $R_t^*(\mu)$.*

*Proof.* The assumption in Eq. (12) ensures that

$$\mathbb{P}\left(\left|R_t(\hat{h}, \mu) - R_t^*(\mu)\right| \geq c\epsilon\right) \leq \mathbb{P}\left(\left|R_t(\hat{h}, 1/\tau) - R_t^*(1/\tau)\right| \geq \epsilon\right)$$

for any $\epsilon$ and $t$. The right hand-side is shown to converge to zero in the proof of Theorem 1 and therefore the left-hand side also converges to zero. $\qquad\square$

A corresponding Theorem 2 also holds for the discounted future loss. Note that Theorems 1 and 2 also hold in a slightly general setting when the measure $\mu^{(\tau)}(\omega)$ is a random variable that depends upon the realization of the stochastic process $\omega \in \Omega$.

## D   An illustrative example of prospective ERM

Suppose we have a stochastic process such that $Z_t \sim p_{(t \bmod T)}$ for some known period $T$, i.e., data is independent across time but not identically distributed, and the loss function $\ell(t, \cdot, \cdot)$ is time-invariant. Scenario 2 is a special case with $T = 2$. Assume that we can find a countable hypothesis class $\mathcal{G}$ that contains the Bayes plug-in estimator for each $p_t$ with $t \in \{1, \ldots, T\}$. Then $\mathcal{H}_T = \{h : h_{t+T} = h_t$ and $h_t \in \mathcal{G} \, \forall t\}$ satisfies Eq. (9), and it is also countable. This implies consistency and uniform concentration of the limsup for sequences in some sub-classes $\{\mathcal{H}_t\}_{t=1}^\infty$ that expands to $\mathcal{H}_T$. Note that even if we do not know the period, we can still implement prospective ERM using the hypothesis class $\cup_{T \in \mathbb{N}} \mathcal{H}_T$; this is a countable set. Prospective ERM is therefore a strong prospective learner if the period $T$ is bounded.

**Remark 4** (**Implementing prospective ERM for periodic processes**). If $\mathcal{G}$ has a finite VC-dimension, choosing $\mathcal{H}_t = \mathcal{H}_T$ for any $t > T$ as the increasing sequence of hypothesis classes in Theorem 1 guarantees the existence of $\lim_{m \to \infty} \frac{1}{m} \sum_{s=1}^m \ell(s, h_s(x_s), y_s)$. We can therefore choose $u_t = t$ in Eq. (7) and thereby select

$$\hat{h} = \arg\min_{h \in \mathcal{H}_t} \frac{1}{t} \sum_{s=1}^t \ell(s, h_s(x_s), y_s)$$

in Eq. (8). For $\mathcal{H}_t = \mathcal{H}_T$ selected above for the periodic process, this is identical to Eq. (5). In other words, implementing prospective ERM for a periodic process boils down to solving $T$ different time-agnostic ERM problems, each using data $\{z_{sT+k}\}_{s=0}^\infty$, $k \in \{1, ..., T\}$. Observe that this is precisely the prospective learner we used for the example in Scenario 2 and Fig. 1.

**Remark 5** (**Sample complexity of prospective ERM for a periodic process**). We can calculate the sample complexity by exploiting the relatedness of the different distributions in the periodic process. First assume $t > T$, i.e., at least one sample from each distribution is available. We again pick $\mathcal{H}_t = \mathcal{H}_T$ for all $t > T$. Let us assume that $\hat{h}_t \in \mathcal{G}$ for all times $t$. Let $C \equiv C(\epsilon/16, \mathcal{G}^T)$ denote the covering number of a hypothesis class of $T$-length sequences of hypotheses $\mathcal{G}^T = \{(h, \ldots, h) : h \in \mathcal{G}\}$ using balls of radius $\epsilon/16$ with respect to loss $\ell$. Then, using Baxter [84, Theorem 4] we can show that, if

$$t \geq \max\left\{\frac{64}{\epsilon^2} \log \frac{4C(\epsilon, \mathcal{G}^T)}{\delta}, \frac{16T}{\epsilon^2}\right\}, \tag{13}$$

then for prospective ERM in Eq. (8) we have

$$\mathbb{E}\left[R_t(\hat{h})\right] \leq \lim_{t \to \infty} \mathbb{E}\left[\inf_{h \in \mathcal{H}_T} R_t(h)\right] + 2\epsilon,$$

with probability at least $1 - \delta$. The sample complexity in Eq. (13) is dominated by the first term in the curly brackets; Baxter [84, Lemma 5] shows that $C(\epsilon, \mathcal{G}^T) \leq (C(\epsilon, \mathcal{G}))^T$. Sample complexity of prospective ERM grows at most linearly with the period $T$, as one would expect.

**Remark 6** (**Exact sample complexity for a periodic binary classification with one-dimensional Gaussian inputs**). Let the period of the stochastic process be $T = 2$ with inputs $X_t \in \mathbb{R}$ and outputs $Y_t \in \{-1, 1\}$. Suppose $Y_t \sim \text{Bernoulli}(0.5)$. The distribution $\mathbb{P}(X_t \mid Y_t = y)$ is a Gaussian with mean $y\mu + \Delta(t \bmod T)$ and variance $\sigma^2$. In words, for even times $t$, the mean of the Gaussians are shifted to the right by $\Delta$. Consider the time-invariant squared error loss $\ell(s, \hat{y}, y) = (\hat{y} - y)^2$. Choose $\mathcal{G} = \{\mathbf{1}_A : A \in \{(-\infty, c), (c, \infty) : c \in \mathbb{R}\}\}$ to be the set of predictors for Fisher's linear discriminant (FLD); the prospective learner selects each element of its hypothesis from $\mathcal{G}$. The calculations in De Silva et al. [61] for FLD can be used to show that if $\mathcal{H}_t = \{(h, h, \ldots) : h \in \mathcal{G}\}$ for all times $t$, then a time-agnostic ERM has risk

$$\mathbb{E}\left[R_t(\hat{h})\right] \to 1/2\left[\Phi\left(\Delta/(2\sigma) - \mu/\sigma\right) + \Phi\left(-\Delta/(2\sigma) - \mu/\sigma\right)\right],$$

where $\Phi$ is the Gauss error function. But if $\mathcal{H}_t = \{(h_1, h_2, h_1, h_2, \dots) : h_1, h_2 \in \mathcal{G}\}$ for all times $t$, then prospective ERM can achieve Bayes risk

$$\mathbb{E}\left[R_t(\hat{h})\right] = \Phi\left(-t\mu/(2\sigma)(t/2(t/2+1))^{-1/2}\right) \to \Phi(-\mu/\sigma) = \mathbb{E}\left[R_t^*\right].$$

**Remark 7** (**Hidden Markov Models (HMMs)**). Suppose $Z$ is sampled from an HMM whose hidden states evolve according to a $k^{\text{th}}$-order time-homogeneous Markov process with a finite state space. Select a hypothesis class $\mathcal{H}_k$ that consists of sequences $h \in \mathcal{H}_k$ such that each $h$ satisfies

$$\forall t \in \mathbb{N} : h_t \in \mathcal{G} \text{ and,}$$
$$\forall t_1, t_2 \in \mathbb{N} : \text{if } h_{t_1+s} = h_{t_2+s} \ \forall s \in \{1\dots, k\}, \text{then } h_{t_1+k+1} = h_{t_2+k+1}.$$

This is the hypothesis class that contains sequences of predictors that depend only on the past $k$ predictors. If we assume, as above, that $\mathcal{G}$ is countable, then so is $\mathcal{H}_k$. And it also satisfies Eq. (9) because of the $k^{\text{th}}$-order Markov property. We can therefore implement prospective ERM using $\mathcal{H}_k$ as the hypothesis class. Observe that in the case when the Markov process underlying the HMM is deterministic, our example models the output from an auto-regressive language model that uses greedy decoding. The length of the context window is $k$, the hidden state of the HMM is the logit at each step (the next hidden state is a deterministic function of the previous $k$ ones), and the output of the HMM $Z_t$ is the next token. Our theory therefore shows that the output of such a model is prospectively learnable if the learner has access to the sequence of tokens.

# E Proofs

## E.1 Proof of Proposition 1

Let $\mathcal{X} = \{-1, 1\}$ and $\mathcal{Y} = \{0, 1\}$. Consider two distributions $P_1$ and $P_2$ (Fig. A.3):

$$P_1(X = x) = P_2(X = x) = \frac{1}{2} \quad \forall x,$$

$$P_1(Y = 1 \mid X = x) = \begin{cases} \theta & \text{if } x = 1 \\ 1 - \theta & \text{if } x = -1, \end{cases}$$

$$P_2(Y = 1 \mid X = x) = \begin{cases} 1 - \theta & \text{if } x = 1 \\ \theta & \text{if } x = -1, \end{cases}$$

In other words, the inputs have the same marginals but the labels are flipped between $P_1$ and $P_2$. Consider a stochastic process $Z$ such that $Z_{2t+1} \sim P_1$ and $Z_{2t} \sim P_2$ where $t \in \mathbb{N}$.

Let $\mathcal{G}$ be any hypothesis class and let $\ell(s, \hat{y}, y) = \mathbf{1}(\hat{y} \neq y)$ be the time-invariant zero-one loss. The time-agnostic learner uses a sequence of hypotheses $h \equiv (h_t)$ where $h_t = h_{t'} \ \forall t, t' \in \mathbb{N}$ to make predictions at all times. The future loss is

$$\bar{\ell}_t(h, Z) = \lim_{\tau \to \infty} \frac{1}{2\tau} \sum_{s=t+1}^{t+2\tau} \ell(s, h_s(X_s), Y_s) = R_1(h) + R_2(h) = \frac{1}{2},$$

almost surely; here $R_1(h)$ and $R_2(h)$ are risks on data from distributions $P_1$ and $P_2$ at odd and even times, respectively. The last equation follows from the fact that $R_1(h) = 1 - R_2(h)$ because the labels are flipped. Prospective Bayes risk is zero if the hypothesis class $\mathcal{G}$ contains the Bayes optimal hypotheses for each of the two distributions. The future loss evaluates to $1/2$ for all realizations and so does the prospective risk. The prospective risk of a hypothesis sequence that makes random predictions (zero or one with equal probability at each instant) is also $1/2$. This stochastic process is not weakly prospective learnable.

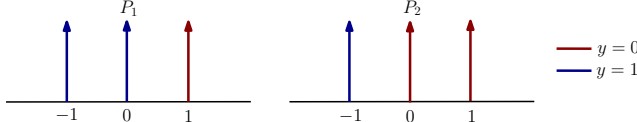

**Figure A.4:** A simple stochastic process that is weakly but not strongly prospectively learnable.

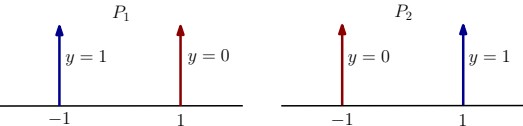

**Figure A.3:** A simple stochastic process that is not weakly prospectively learnable.

Now consider the two distributions shown in Fig. A.4,

$$P_1(X = x) = P_2(X = x) = \frac{1}{3} \quad \forall x,$$

$$P_1(Y = 1 \mid X = x) = \begin{cases} \theta & \text{if } x \leq 0 \\ 1 - \theta & \text{if } x = 1, \end{cases}$$

$$P_2(Y = 1 \mid X = x) = \begin{cases} 1 - \theta & \text{if } x \geq 0 \\ \theta & \text{if } x = -1. \end{cases}$$

Inputs are supported on the set $\{-1, 0, 1\}$ this time. Again consider a stochastic process $Z$ such that $Z_{2t+1} \sim P_1$ and $Z_{2t} \sim P_2$ for $t \in \mathbb{N}$. For a time-agnostic learner, since its hypothesis $h$ at each time step has to predict incorrectly at $x = 0$, we have $R_1(h) + R_2(h) \geq \frac{1}{3}$. The future loss is

$$\bar{\ell}_t(h, Z) = \lim_{\tau \to \infty} \frac{1}{2\tau} \sum_{s=t+1}^{t+2\tau} \ell(s, h(X_s), Y_s) = R_1(h) + R_2(h) \geq \frac{1}{3}.$$

almost surely. It follows that the prospective risk $R_t(h) \geq \frac{1}{3}$ for any hypothesis. Prospective Bayes risk is again zero and therefore this stochastic process is not strongly prospectively learnable. It is however weakly learnable.

A hypothesis that predicts $\hat{y} = \pm 1$ with equal probability has $R_t^0 = 0.5$. If the data contains samples for $x \in \{-1, 1\}$, ERM will select a hypothesis that minimizes the empirical risk which necessitates that $h(1) = 0$ and $h(-1) = 1$. Therefore $R_1(h) + R_2(h) \leq \frac{1}{3} + \epsilon$, since $h$ predicts correctly at $x = \pm 1$, and incorrectly at $x = 0$ exactly one of the two distributions. The constant $\epsilon$ can be chosen to be $\propto t^{-1/2}$ after receiving data from $t$ timesteps. The probability with which we do not get samples at $x = 1$ or at $x = -1$, is $2 \times 3^{-t}$. Therefore the probability that $R_1(h) + R_2(h) \leq \frac{1}{3} + \epsilon$ is at least $1 - 3^{-t+1}$ after $t$ time steps. This learner is therefore better than the chance learner whose risk is $R_t^0$ and it is a weak prospective learner. This shows that there exist stochastic processes that are weakly prospective learnable using time-agnostic ERM but not strongly.

### E.2 Proof of Theorem 1

We first show that for each $Z \in \mathcal{Z}$, if Eqs. (6) and (7) holds, then the risk of estimator in Eq. (8) converges in probability to the Bayes optimal, i.e.

$$\mathbb{P}\left( \left| R_t(\hat{h}^{(t)}) - R_t^* \right| < \epsilon \right) \to 1.$$

By taking

$$t = \max \left\{ t : \mathbb{P}\left( \left| R_t(\hat{h}^{(t)}) - R_t^* \right| < \epsilon \right) \geq 1 - \delta \quad \forall t' \geq t, Z \in \mathcal{Z} \right\}$$

we can trivally show that the strong prospecitve learnability of estimator in Eq. (8) holds over the family $\mathcal{Z}$.

We first state a lemma that gives a choice of a random hypothesis $h^{(t)} \in \sigma(Z_{\leq t})$ converging to the Bayes optimal risk under the consistency assumption. First we define the shorthand

$$e_m(h) \equiv \frac{1}{m} \sum_{s=1}^{m} \ell(s, h_s(X_s), Y_s).$$

**Lemma 3.** For a stochastic process $Z$, for an increasing sequence of hypothesis $\mathcal{H}_1 \subseteq \mathcal{H}_2 \subseteq \ldots$ such that the consistency condition in Eq. (6) is satisfied, for any $u_t$ satisfying $u_t \leq t$, $u_t \to \infty$, there exists $h^{(t)} \in \sigma(Z_{\leq t})$, a $\mathcal{H}_t$-valued random variable, such that

$$\mathbb{E}\left[ \sup_{u_t \leq m \leq \infty} e_m(h^{(t)}) \mid Z_{\leq t} \right] - R_t^* \to 0 \tag{14}$$

almost surely.

*Proof.* By Eq. (6), there exists $h^{(t)} \in \sigma(Z_{\leq t})$, a $\mathcal{H}_t$-valued random variable such that

$$\lim_{t \to \infty} \mathbb{E}\left[ R_t(h^{(t)}) - R_t^* \right] = 0$$

Here, $h^{(t)} \in \sigma(Z_{\leq t})$ means that $h^{(t)}$ is constant on the set $\{Z_{\leq t} = z_{\leq t}\}$. By the assumption on $h^{(t)}$, we have

$$R_t(h^{(t)}) \geq \inf_{h \in \mathcal{H}_t} R_t(h) \geq R_t^*$$

almost surely. We can choose a sub-sequence $\{j_k\}$, such that $\mathbb{E}\left[R_{j_k}(h^{(j_k)}) - R_{j_k}^*\right] \leq 4^{-k}$. For all random variables $h^{(t)}$ satisfying the above assumption, the bounded convergence theorem implies that

$$\mathbb{E}[R_t(h^{(t)}) - R_t^*] = \mathbb{E}\left[\mathbb{E}\left[\limsup_{m \to \infty} e_m(h^{(t)}) \mid Z_{\leq t}\right] - R_t^*\right]$$

$$= \mathbb{E}\left[\mathbb{E}\left[\lim_{i \to \infty} \sup_{u_i \leq m \leq \infty} e_m(h^{(t)}) \mid Z_{\leq t}\right]\right] - \mathbb{E}\left[R_t^*\right]$$

$$= \lim_{i \to \infty} \mathbb{E}\left[\mathbb{E}\left[\sup_{u_i \leq m \leq \infty} e_m(h^{(t)}) \mid Z_{\leq t}\right] - R_t^*\right]$$

In particular, this implies that for any integer $k$ there exists an integer $i_k$ such that

$$\mathbb{E}\left[\mathbb{E}\left[\sup_{u_{i_k} \leq m \leq \infty} e_m(h^{(j_k)}) \mid Z_{\leq j_k}\right] - R_{j_k}^*\right] \leq \mathbb{E}\left[R_{j_k}(h^{(j_k)}) - R_{j_k}^*\right] + 4^{-k} \leq 2 \times 4^{-k}.$$

By the definition of limsup, we have

$$\mathbb{E}\left[\sup_{u_i \leq m \leq \infty} e_m(h^{(t)}) \mid Z_{\leq t}\right] \geq \mathbb{E}\left[\bar{\ell}_t(h^{(t)}, Z) \mid Z_{\leq t}\right] \geq R_t^*$$

and therefore we can use Markov's inequality to get

$$\sum_{k=0}^{\infty} \mathbb{P}\left(\mathbb{E}\left[\sup_{u_{i_k} \leq m \leq \infty} e_m(h^{(j_k)}) \mid Z_{\leq j_k}\right] - R_{j_k}^* > 2^{(1/2)-k}\right)$$

$$\leq \sum_{k=0}^{\infty} \frac{1}{2^{(1/2)-k}} \mathbb{E}\left[\mathbb{E}\left[\sup_{u_{i_k} \leq m \leq \infty} e_m(h^{(j_k)}) \mid Z_{\leq j_k}\right] - R_{j_k}^*\right]$$

$$\leq \sum_{k=0}^{\infty} \frac{2}{2^{(1/2)-k}} 4^{-k} < \infty.$$

Therefore, by Borel-Cantelli lemma, with probability one, there exists a (random) integer $k_0$ such that for all $k \geq k_0$,

$$\mathbb{E}\left[\sup_{u_{i_k} \leq m \leq \infty} e_m(h^{(j_k)}) \mid Z_{\leq j_k}\right] - R_{j_k}^* \leq 2^{(1/2)-k}$$

We will now scale $i_k, j_k$ by some integer $k_t$ such that $i_{k_t}, j_{k_t} \leq t$. This ensures that $u_{i_{k_t}} \leq u_t$ and $\mathcal{H}_{i_{k_t}} \subseteq \mathcal{H}_t$. This scaling is necessary to relate the "empirical estimate" of the $\limsup$ of $\hat{h}$ to the actual $\limsup$ of $h^{(t)}$. To that end, define

$$k_t = \max\{k \in \mathbb{N} \cup \{0\} : \max\{i_k, j_k\} \leq t\}.$$

Since $i_k \leq \infty$ and $j_k \leq \infty$, we also have $\lim_{t \to \infty} k_t = \infty$. Let $\alpha_t = 2^{(1/2)-k_t}$ and notice that $\alpha_t \to 0$. We can construct an integer-valued random variable $t_0$ such that for all $t \geq t_0$, we have $k_t \geq k_0$, and therefore

$$\mathbb{E}\left[\sup_{u_{i_{k_t}} \leq m \leq \infty} e_m(h^{(j_{k_t})}) \mid Z_{\leq j_{k_t}}\right] - R_{j_{k_t}}^* \leq \alpha_t.$$

Now choose $h^{(t)} = h^{(j_{k_t})}$ for every $t \in \mathbb{N}$. Since $j_{k_t} \leq t$, we have $\mathcal{H}_{j_{k_t}} \subseteq \mathcal{H}_t$ and $\sigma(Z_{\leq j_{k_t}}) \subseteq \sigma(Z_{\leq t})$. This implies that $h^{(j_{k_t})}$ is an $\mathcal{H}_t$-valued random variable and $h^{(j_{k_t})} \in \sigma(Z_{\leq t})$. Also, since $i_{k_t} \leq t$ and $\{u_t\}_{t=1}^{\infty}$ is non-decreasing, we have $u_{i_{k_t}} \leq u_t$ for all $t \in \mathbb{N}$. Hence, with probability one, $\forall t \geq t_0$,

$$\mathbb{E}\left[\sup_{u_t \leq m \leq \infty} e_m(h^{(t)}) \mid Z_{\leq j_{k_t}}\right] - R_{j_{k_t}}^* \leq \mathbb{E}\left[\sup_{u_{i_{k_t}} \leq m \leq \infty} e_m(h^{(t)}) \mid Z_{\leq j_{k_t}}\right] - R_{j_{k_t}}^* \leq \alpha_t$$

Since $\alpha_t \to 0$, we have

$$\mathbb{E}\left[\sup_{u_t \leq m \leq \infty} e_m(h^{(t)}) \mid Z_{\leq j_{k_t}}\right] - R^*_{j_{k_t}} \to 0 \text{ a.s.}$$

Again using the bounded convergence theorem,

$$\mathbb{E}\left[\sup_{u_t \leq m \leq \infty} e_m(h^{(t)}) \mid Z_{\leq t}\right] - R^*_t \to 0 \text{ a.s.}$$

$\square$

Now we continue the proof of Theorem 1 by exploiting the convergence of the empirical $\limsup$ to the true $\limsup$. We construct a sub-sequence of integers $i_t$ such that $\gamma_{i_t}$ in Eq. (7) decays exponentially. Markov's inequality implies that

$$\sum_{t=0}^{\infty} \mathbb{P}\left(\max_{h \in \mathcal{H}_{i_t}} \left|\bar{\ell}_t(h, Z) - \max_{u_{i_t} \leq m \leq i_t} e_m(h)\right| > \sqrt{\gamma_{i_t}}\right)$$

$$\leq \sum_{t=0}^{\infty} \frac{1}{\sqrt{\gamma_{i_t}}} \mathbb{E}\left[\max_{h \in \mathcal{H}_{i_t}} \left|\bar{\ell}_t(h, Z) - \max_{u_{i_t} \leq m \leq i_t} e_m(h)\right|\right] \leq \sum_{t=0}^{\infty} \sqrt{\gamma_{i_t}} < \infty.$$

By the Borel-Cantelli lemma, with probability one, there exists $t_1 \in \mathbb{N}$, random, and $t_1 \in \mathcal{F}$, such that $\forall t \geq t_1$,

$$\max_{h \in \mathcal{H}_{i_t}} \left|\bar{\ell}_t(h, Z) - \max_{u_{i_t} \leq m \leq i_t} e_m(h)\right| \leq \sqrt{\gamma_{i_t}}.$$

Let $j_t = \max\{t' : i_{t'} \leq t\}$, then we have $i_{j_t} \to \infty$. Since $i_{j_t} \leq t$, we have

$$\bar{\ell}_t(h, Z) - \max_{u_{i_{j_t}} \leq m \leq t} e_m(h) \leq \bar{\ell}_t(h, Z) - \max_{u_{i_{j_t}} \leq m \leq i_{j_t}} e_m(h).$$

Construct a random variable $t_2$ such that $\forall t \geq t_2$, we have $j_t \geq t_1$. Hence, we have for all $t \geq t_2$,

$$\max_{h \in \mathcal{H}_{i_{j_t}}} \left\{\bar{\ell}_t(h, Z) - \max_{u_{i_{j_t}} \leq m \leq t} e_m(h)\right\}$$

$$\leq \max_{h \in \mathcal{H}_{i_{j_t}}} \left|\bar{\ell}_t(h, Z) - \max_{u_{i_{j_t}} \leq m \leq i_{j_t}} e_m(h)\right|$$

$$\leq \sqrt{\gamma_{i_{j_t}}}.$$

Let $i_t = i_{j_t}$ and notice that since we have $i_t \to \infty$ we also have $i_t \leq t$. This gives a sub-sequence that depends only on $\gamma_t$. Then, with probability one, $\forall t \geq t_2$

$$\max_{h \in \mathcal{H}_{i_t}} \left\{\bar{\ell}_t(h, Z) - \max_{u_{i_t} \leq m \leq t} e_m(h)\right\} \leq \sqrt{\gamma_{i_t}}.$$

Let $\{h^{(t)}\}_{t=1}^{\infty}$, where $h^{(t)} \in \sigma(Z_{\leq t})$ is a $\mathcal{H}_{i_t}$-valued random variable chosen as in Lemma 3 with $\mathcal{H}_t$ chosen to be $\mathcal{H}_{i_t}$ and $u_t$ chosen to be $u_{i_t}$. Since $\hat{h}^{(t)} \in \sigma(Z_{\leq t})$, with probability one, for $t \geq t_2$,

$$\bar{\ell}_t(\hat{h}^{(t)}, Z) \leq \max_{u_{i_t} \leq m \leq t} e_m(\hat{h}^{(t)}) + \sqrt{\gamma_{i_t}}$$

$$\leq \max_{u_{i_t} \leq m \leq t} e_m(h^{(t)}) + \sqrt{\gamma_{i_t}}$$

$$\leq \sup_{u_{i_t} \leq m \leq \infty} e_m(h^{(t)}) + \sqrt{\gamma_{i_t}}.$$

Hence,

$$\mathbb{E}\left[\bar{\ell}_t(\hat{h}^{(t)}, Z) \mid Z_{\leq t}\right] - \mathbb{E}\left[\sup_{u_{i_t} \leq m \leq \infty} e_m(h^{(t)}) \mid Z_{\leq t}\right] \to 0$$

almost surely. By Lemma 3, we have,

$$\mathbb{E}\left[\bar{\ell}(\hat{h}^{(t)}, Z) \mid Z_{\leq t}\right] - R^*_t \to 0$$

almost surely. Hence, by the bounded convergence theorem,

$$0 = \mathbb{E}\left[\lim_{t\to\infty}\left(\mathbb{E}\left[\bar{\ell}(\hat{h}^{(t)}, Z) \mid Z_{\leq t}\right] - R_t^*\right)\right] = \lim_{t\to\infty}\mathbb{E}\left[\mathbb{E}\left[\bar{\ell}(\hat{h}^{(t)}, Z) \mid Z_{\leq t}\right] - R_t^*\right]$$

which implies that

$$\mathbb{P}\left(\left|R_t(\hat{h}^{(t)}) - R_t^*\right| \geq \delta\right) \leq \frac{1}{\delta}\mathbb{E}\left[\mathbb{E}\left[\bar{\ell}(\hat{h}^{(t)}, Z) \mid Z_{\leq t}\right] - R_t^*\right] \to 0.$$

We have therefore proved that $\hat{h}^{(t)}$ is a strong prospective learner.

### E.3 Proof of Theorem 2

This construction follows closely with Hanneke [85, Section 4], and we omit some details here. For finite class $\mathcal{H}$ of sequence of hypothesis, we give a possible choice of $u_t$ with

$$\lim_{t\to\infty}\mathbb{E}\left[\sup_{t'\geq t}\max_{h\in\mathcal{H}}\left|\bar{\ell}_t(h, Z) - \max_{u_t\leq m\leq t}e_m(h)\right|\right] = 0. \tag{15}$$

for all process $Z$ in the finite family $\mathcal{Z}$. For a data sequence $\mathbf{z} = \{z_s = (x_s, y_s)\}_{s=0}^{\infty}$ and a hypothesis sequence $h \in \mathcal{H}$, we define

$$t_u^h(\mathbf{z}) = \min\left\{t \in \mathbb{N} : t \geq u, \forall t' \geq t \sup_{u\leq m\leq\infty}e_m(h) \leq \max_{u\leq m\leq t'}e_m(h) + 2^{-u}\right\},$$

and

$$u_t^h(\mathbf{z}) = \max\{u \in \{1, \ldots, t\} : t \geq t_u^h(\mathbf{z})\},$$
$$u_t^{\mathcal{H}}(\mathbf{z}) = \min_{h\in\mathcal{H}}u_t^h(\mathbf{z});$$

note that $\mathcal{H}$ is finite and therefore the minimum exists. For the stochastic process $Z$, let

$$u_t^{\mathcal{H}}(\delta, Z) = \max\left\{u \in \{1, \ldots, t\} : \mathbb{P}_{\mathbf{z}\sim Z}(u_t^{\mathcal{H}}(\mathbf{z}) \geq u) = 1 - \delta\right\},$$

$$u_t(Z) = \max\left\{s \in \mathbb{N} \cup \{0\} : u_t^{\mathcal{H}}(2^{-u}, Z) \geq u\right\},$$

$$u_t = \min\left\{u_t(Z) : Z \in \mathcal{Z}\right\}$$

Since $\mathcal{Z}$ is finite, we have $u_t \to \infty$. And we also have $\forall Z \in \mathcal{Z}$,

$$\mathbb{P}\left(u_t^{\mathcal{H}}(Z) \geq u_t\right) \geq 1 - 2^{-u_t},$$

By Borel-Cantelli Lemma, this construction gives a sequence $u_t$ s.t. Eq. (15) holds, i.e. $\forall Z \in \mathcal{Z}$,

Suppose $\{\mathcal{H}_t\}_{t=1}^{\infty}$ is a sequence of non-empty finite sets of hypothesis sequences, and $\{\gamma_t\}_{t=1}^{\infty}$ is a sequence in $(0, \infty)$ with $\gamma_1 \geq 1$. Then for any finite family $\mathcal{Z}$, we can extend this construction to get a choice of $u_t$, $\mathcal{H}_t$ and $\gamma_t$ such that Eq. (7) holds

$$\mathbb{E}\left[\max_{h\in\mathcal{H}_t}\left|\bar{\ell}_t(h, Z) - \max_{u_t\leq m\leq t}e_m(h)\right|\right] \leq \gamma_t.$$

Since we have a $u_t$ for a given hypothesis class $\mathcal{H}$, for each $i \in \mathbb{N}$, we can construct a sequence $\{u_{i,t}\}_{t=1}^{\infty}$ such that $\lim_{t\to\infty}u_{i,t} = \infty$, $u_{i,t} < t$ and $\forall Z \in \mathcal{Z}$,

$$\lim_{t\to\infty}\mathbb{E}\left[\sup_{t'\geq t}\max_{h\in\mathcal{H}_i}\left|\bar{\ell}_t(h, Z) - \max_{u_{i,t}\leq m<n'}e_m(h)\right|\right] = 0.$$

Now let

$$j_t = \max\left\{i \in \{1, \ldots, t\} : \forall i' \leq i, \sup_{t''\geq t}\mathbb{E}\left[\sup_{t'\geq t''}\max_{h\in\mathcal{H}_{i'}}\left|\bar{\ell}_t(h, Z) - \max_{u_{i',t''}\leq m\leq t'}e_m(h)\right|\right] \leq \gamma_{i'}\forall Z \in \mathcal{Z}\right\}$$

and

$$t_i = \min\left\{t : j_t \geq i, u_{i,t} > u_{i-1,n_{i-1}}\right\}.$$

If $i_t = \max\{i : t_i < t\}$, then we have $i_t \to \infty$, and for $u_i = u_{i,t_i}$ we have

$$\mathbb{E}\left[\max_{h\in\mathcal{H}_{i_t}}\left|\bar{\ell}_t(h, Z) - \max_{u_{i_t}\leq m\leq t}e_m(h)\right|\right] \leq \gamma_{i_t}.$$

Since $\mathcal{H}$ is countable, we can choose a sequence of finite hypothesis classes $\mathcal{H}_t$ such that $\cup_{t\in\mathbb{N}}\mathcal{H}_t = \mathcal{H}$. By choosing $u_t = u_{i_t}$, with $\mathcal{H}_t = \mathcal{H}_{i_t}$, and $\gamma_t = \gamma_{i_t}$, we now have a possible choice for $u_t$, $\mathcal{H}_t$ and $\gamma_t$ in Theorem 1.

# F    Experimental Setup

## F.1    Training and evaluation

**Training setup.**    Each learner receives a $t$-length sequence of samples $z_{\leq t}$ drawn from the stochastic process, as the training data. Upon training, the learner is expected to make predictions on future samples that correspond to times $t' > t$ up to a fixed horizon $T$. At each future time $t'$, we do not train (modify the weights) using samples after time $t$ (because we do not have them, but we will make predictions on these samples). Given samples $z_{\leq t}$, a time-aware hypothesis class minimizes the empirical prospective risk

$$\hat{\ell}_t(h, Z) = \frac{1}{t} \sum_{s=1}^{t} \ell(s, h_s(x_s), y_s);$$

For an MLP or CNN, $h_s$ corresponds to a network that takes time $s$ as input.

**Hyper-parameters**    All the networks are trained using stochastic gradient descent (SGD) with Nesterov's momentum and cosine-annealed learning rate. The networks are trained at a learning rate of $0.1$ for the synthetic tasks, and learning rate of $0.01$ for MNIST and CIFAR. The weight-decay is set to $1 \times 10^{-5}$. The images from MNIST and CIFAR-10 are normalized to have mean $0.5$ and standard deviation $0.25$. The models were trained for 100 epochs, which is many epochs after achieving a training accuracy of 1.

**Evaluation**    We estimate the prospective risk of each learner using a Monte Carlo estimate. For a given training dataset $z_{\leq t}$, we estimate a sequence predictors $h \equiv (h_t)$ which we use to make predictions on future samples. We wish to approximate the prospective risk (Equation (2)) for the estimated sequence of predictors. We do so, for a single future realization $z_{>t}$ of this process, which yields the estimate

$$\hat{R}_t(h) = \frac{1}{(T-t)} \sum_{s=t+1}^{T} \ell\left(s, h_s(x_s^j), y_s^j\right).$$

In our experiments, $T = 50{,}000$ for CIFAR-10 and MNIST while $T = 10{,}000$ for the synthetic data experiments. For a single learning algorithm, we compute the empirical prospective risk at 15-40 different time steps which results in a significant number of GPU hours in order to plot the learning curves. For every time step, we compute the mean and standard deviation of the empirical prospective risk using 5 random seeds.

## F.2    Architectural Details

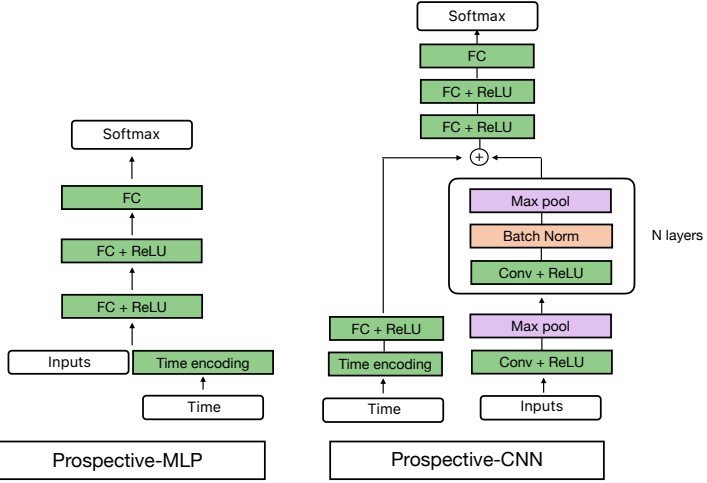

**Figure A.5:** Schematic illustration prospective-MLP and prospective-CNN.

We considered the following architecture choices for the time-agnostic restropective algorithms like ERM that ignore time and the ordering associated with the samples in $z_{\leq t}$.

**Retrospectron-MLP/CNN.**    A multi-layer perceptron (MLP) with two hidden layers with 256 units is used for the synthetic tasks and the MNIST task. For CIFAR-10, we use a a small convolutional network with 0.12M parameters. It comprises of 3 convolution layers (kernel size 3 and 80 filters) interleaved with max-pooling, ReLU, batch-norm layers, with a fully-connected classifier layer.

**Prospective ERM with MLP and CNNs.** In order to incorporate time into the hypothesis class, we consider an embedding function $\varphi : \mathbb{R} \to \mathbb{R}^d$ that takes raw time as an input and returns a $d$-dimensional vector denoted as the time-embedding. In our experiments, we define $\varphi : \mathbb{R} \to \mathbb{R}^d$ as a function that maps

$$t \mapsto (\sin(\omega_1 t), \dots, \sin(\omega_{d/2} t), \cos(\omega_1 t), \dots, \cos(\omega_{d/2} t)),$$

where, $\omega_i = \pi/i$, $i = 1, \dots, d/2$ to the be the collection of angular frequencies. We briefly discuss the rationale for this choice in Figure A.7. In our experiments, we use $d = 50$.

We make our classifiers a function of time by including time $t$ as an input the neural network. This allows the network to vary its hypothesis over time. For MLPs, we concatenate the input with its corresponding time-embedding $\varphi(t)$ which is fed as input. For the CNN (see Figure A.5), we add the time-embedding to the output of the convolutional layers instead of concatenating it to the inputs. We also tried concatenating the time-encoding to the inputs of the CNN but found that it performed poorly in both scenarios 2 and 3 (see Figure A.6).

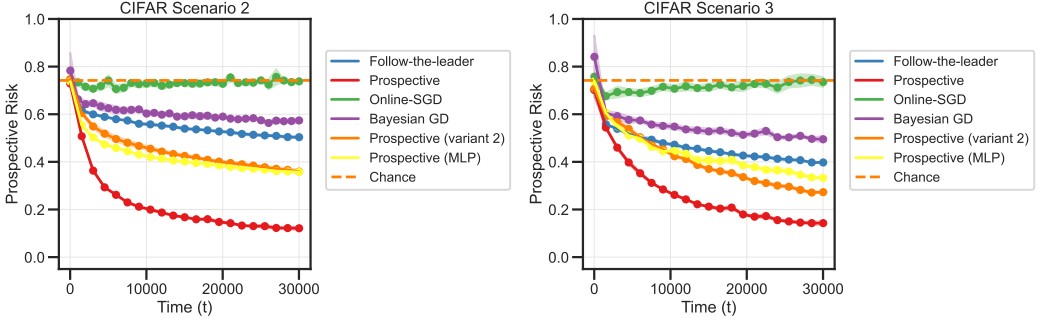

**Figure A.6:** Prospective risk of the CNN architecture on CIFAR-10 for scenarios 2 and 3. The performance of the CNN architecture is significantly worse when the time-embedding is concatenated to the input (variant 2).

**Frequencies for embedding time** In the original Transformer architecture, Vaswani et al. [55] use a position-embedding using the frequencies $\omega_i = 1/10000^{2i/d}$ $i = 1, \dots, d/2$. There are two key differences: (1) We use the absolute time $t$ instead of the relative position, (2) We use the angular frequencies $2\pi/i$. In Figure A.7 *(right)*, we illustrate the time-embeddings when we use the two different choices for angular frequencies. For $d = 128$, we find that the frequencies from Vaswani et al. [55] result in slowly changing features which makes it less suitable for our task, i.e., many of the dimensions are constant over time which makes many of the dimensions uninformative for the task. In our experiments, we found out that MLPs and CNNs that use the frequencies from Vaswani et al. [55] perform poorly on the MNIST task for Scenario 2 and 3.

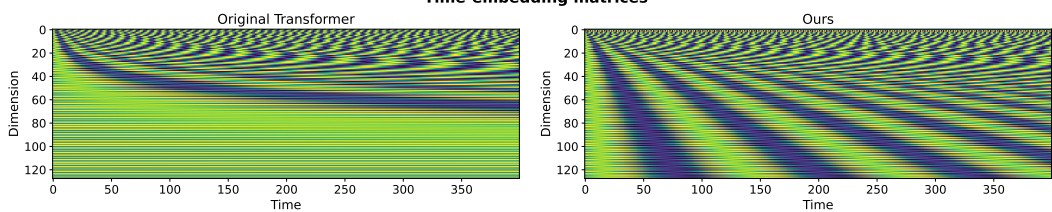

**Figure A.7:** The time-embeddings computed using (1) frequencies from Vaswani et al. [55] *(left)*, and (2) the frequencies from proposed in our work *(right)*.

# G    Additional experiments for Scenario 3

We conduct more experiments on prospective learning problems in addition to the ones in Section 6.2.

## G.1    Markov chain with periodic resets

**Dataset and Tasks.** For synthetic data, we consider the 2 binary classification problems described in Section 6.1. For CIFAR-10 and MNIST, we consider 2 tasks corresponding to the classes 1-5, and the classes 1-5 but with each class $y$ relabeled to $(y + 1) \pmod 5$. Using these tasks, we construct Scenario 3 problems corresponding to a stochastic process which is a hidden Markov model on two states. The tasks are governed by a Markov process with transition matrix $P(S_{t+1} = k \mid S_t = k) = 0.1$, where $S_t$ is the task at time $t$. Additionally after every 10 time-steps, the state of the Markov chain is reset to the first task. This ensures that the stochastic process does not have a stationary distribution.

Similar to the previous experiments, for each problem, we generate a sequence of 50,000 samples. Learners are trained on data from the first $t$ time steps $(z_{\leq t})$ and prospective risk is computed using samples from the remaining time steps.

**Learners and hypothesis classes.**  For this scenario, we conduct experiments using follow-the-leader and prospective ERM. Both methods use MLPs for synthetic and MNIST tasks, and a CNN for the CIFAR-10 task. Note that prospective ERM uses an embedding of time as input in addition to the datum. Training and evaluation setup is identical to that of Scenario 2.

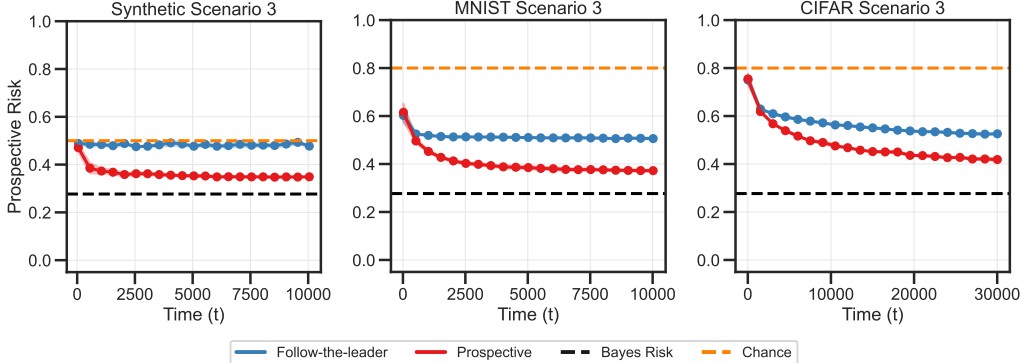

**Figure A.8: Prospective ERM can achieve good prospective risk in Scenario 3.** We plot the prospective risk across 5 random seeds (which govern the sequence of samples and the weight initialization of the neural networks). In all three cases, the risk of prospective ERM approaches Bayes risk while Follow-the-Leader does not achieve a low prospective risk. Bayes risk for MNIST and CIFAR-10 problems is calculated by assuming that Bayes risk on individual tasks is zero.

As Figure A.8 shows, **prospective ERM can also prospectively learn problems in Scenario 3 when data is neither independent nor identically distributed**.

### G.2   Stationary Markov chain

**Dataset and Tasks.**  For synthetic data, we consider the 2 binary classification problems described in Section 6.1. For CIFAR-10 and MNIST, we consider 2 tasks corresponding to the classes 1-5, and the classes 1-5 but with each class $y$ relabeled to $(y+1) \mod 5$. Using these tasks, we construct Scenario 3 problems corresponding to a stochastic process which is a hidden Markov model on 2 states. The tasks are governed by a Markov process with transition matrix $P(S_{t+1} = k \mid S_t = k) = 0.1$, where $S_t$ is the task at time $t$. Unlike the previous subsection (Figure A.8), in this experiment, the Markov chain equilibrates to the stationary distribution. Similar to the previous experiments, for each problem, we generate a sequence of 50,000 samples. Learners are trained on data from the first $t$ time steps $(z_{\leq t})$ and prospective risk is computed using samples from the remaining time steps.

**Learners and hypothesis classes.**  For this scenario, we conduct experiments using follow-the-leader and prospective ERM. Both methods use MLPs for the synthetic and MNIST tasks, and a CNN for the CIFAR-10 task. Note that prospective ERM uses an embedding of time as input in addition to the datum. Training is identical to that of Scenario 2. For evaluation, we compute the empirical prospective risk in Fig. A.9 and empirical discounted prospective risk in Fig. A.10.

## H   Large language models may not be good prospective learners

It is an interesting question whether LLMs which are trained using auto-regressive likelihoods with Transformer-based architectures can do prospective learning. To study this, we used LLama-7B [86] and Gemma-7B [87] to evaluate the prospective risk for Scenarios 1 to 3. The prompt contains a few samples from the stochastic process (sub-sequences of $(Y_t)$ consisting of 0s and 1s) and an English language description of the family of stochastic processes that generated the samples. The LLM is tasked with completing the prompt with the next 20 most likely sequence of samples.

**Selecting the appropriate prompt** LLMs can be brittle and are known to generate different completions depending on if the prompt was in English, Thai or Swahili [88]. This makes it difficult to evaluate prospective learning in LLMs. Therefore, in our experiments, we do not describe prospective risk or other details about prospective learnability in the prompt. We simply describe the data generating process and some samples from this process in the prompt and prompt the model to generate the most likely completion. The prompts are described in detail in Appendix H.1; we also experimented with a few variants of these prompts.

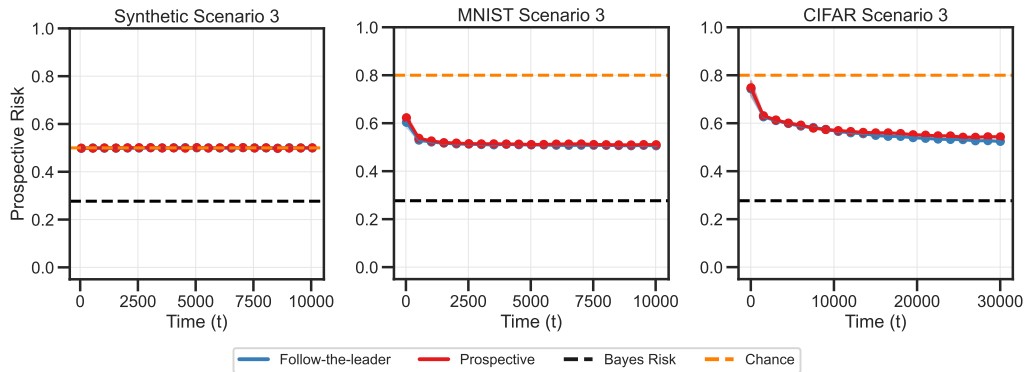

**Figure A.9: For a task defined on a stationary Markov process, the Bayes risk is trivial and can be achieved by a hypothesis that doesn't change over time.** We plot the prospective risk across 5 random seeds (which govern the sequence of samples and the weight initialization of the neural networks). In all three cases, both follow-the-leader and prospective ERM approach the Bayes risk. The stationary distribution has an equal probability of seeing either task and a fixed hypothesis can achieve Bayes risk on this problem.

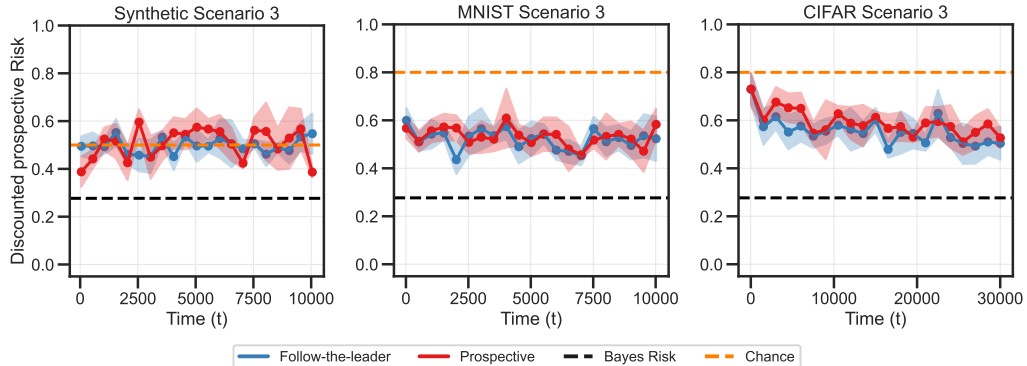

**Figure A.10: Both prospective ERM and follow-the-leader achieve similar discounted prospective risks (with discount factor 0.95).** We plot the discounted prospective risk across 5 random seeds. Both follow-the-leader and prospective ERM achieve similar discounted risks. Note that the error bars are larger since the risk is computed over fewer samples, i.e., the discount factor reduces the effective number of test data points.

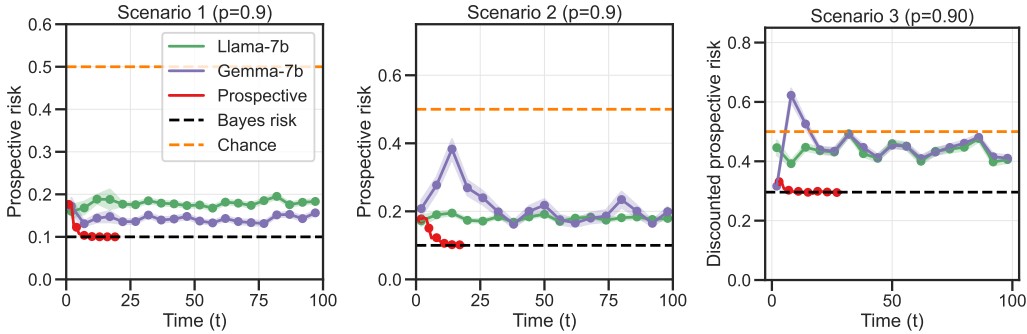

**Figure A.11:** The prospective risk of LLMs when evaluated on the three scenarios, when averaged over draws of the training data. The LLM does not improve with more data unlike a prospective MLE-learner. This suggests that LLMs are incapable of prospection.

We use greedy decoding to generate a sequence of tokens, i.e., the token with the highest probability is sampled at every step. We vary the number of time-steps in the prompt from 1 to 100 which corresponds to the amount of training data. For a particular value of time $t$, we generate 20 more tokens and compute (an estimate of) the prospective risk of this completion; this is the test data. We report the prospective risk computed on 100 different realizations of the stochastic process, i.e., each point in Fig. A.11 is the prospective risk on the next 20 samples, averaged over 100 realizations of the training data. In Fig. A.11, we find that LLMs do not obtain better prospective risk with more

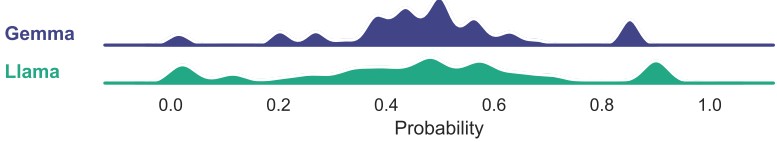

**Figure A.12:** We prompt LLMs to generate the outcomes of 10 Bernoulli trials with $p = 0.75$. We plot the probability of generating token 1 over all possible sequences of 10 Bernoulli trials and find that the outcomes are generated with probabilities that range from 0 to 1 with an average of 0.5. Ideally, the token 1 should should always be generated with $p = 0.75$, i.e., the LLMs cannot simulate outcomes of a Bernoulli distribution.

samples, i.e., Llama-7B and Gemma-7B do not seem to be doing prospective learning. It is quite surprising that they do not achieve Bayes risk even on independent and identically distributed data. We note that these experiments do not definitively answer whether LLMs can learn prospectively.

**Can LLMs even generate outcomes of a sequence of Bernoulli trials?** We prompted an LLM to generate a sequence of 0s and 1s sampled from a Bernoulli distribution with probability $p = 0.75$. We then plot the probability of generating each 0 or 1, for all sequences of length 10 in Fig. A.12. Ideally, the strip plot would be concentrated around 0.25 for 0 and 0.75 for 1, i.e., 0s should be generated with frequency close to 0.25. However, we find this is not the case and LLMs seem incapable of even generating a sequence of Bernoulli trials. This provides some context to the results discussed above. LLMs do not seem to be doing prospective learning, but they cannot even sample from a Bernoulli distribution under these experimental conditions.[13]

### H.1 Prompts for testing prospective learning in LLMs

We use the following 3 prompts to generate a sequence of predictions using in LLama-7B and Gemma-7B. We found that the LLMs always generated a sequence of 0s and 1s and we did not need to post-process the response or change how the tokens were sampled. We generate 20 samples using greedy decoding; the language models are executed with the weights in 16-bit precision. We tried a few different variants for providing prompts to the LLM, e.g., by adding spaces between the 1s and 0s, the results are qualitatively similar.

---

> **Scenario 1**
>
> Consider the following sequence of outcomes generated from a single Bernoulli distribution.
>
> 1111010111111011111111111111
>
> The next 20 most likely outcomes are:

---

> **Scenario 2**
>
> Consider the following sequence of outcomes generated by two Bernoulli distributions, where all even outcomes are generated by a Bernoulli distribution with parameter 'p' and odd outcomes are generated from a Bernoulli distribution with parameter '1-p'.
>
> 10101010101010101010101000101010101010101
>
> The next 20 most likely sequence of outcomes are:

---

[13] Responses of ChatGPT-turbo and GPT-4o were more verbose compared to those of Llama-7B and Gemma-7B. ChatGPT responded correctly to Scenario 1, perhaps as a result of using a scratchpad [89, 90] for generating the results of intermediate steps of the algorithm. But it did not achieve a small prospective risk for Scenarios 2 and 3. Gemini and GPT-4o refused to give a complete response to Scenario 3 and only outlined the sequence of steps, albeit correctly.

Consider the following sequence of states generated by a Markov process with 2 states (0, 1):

10101101010100101010

The next 20 most likely outcomes are:

To make the LLM generate a sequence of Bernoulli trials with probability 0.75, we used the following prompt.

Generate outcomes of 10 Bernoulli trials where 0 is generated with probability 0.25 and 1 with probability 0.75

