# OpenReview forum: "Prospective Learning: Learning for a Dynamic Future"
_NeurIPS.cc/2024/Conference — NeurIPS 2024 poster_

### Official Review · Reviewer_Zv5g · 2024-07-10

**Soundness:** 3
**Presentation:** 2
**Contribution:** 3
**Rating:** 7
**Confidence:** 3

**Summary:**

In this work, the authors formalize the notion of “prospective learning”, which considers that the data to be learned are sampled from a stochastic process, rather than being sampled from a fixed distribution. Perspective learning considers time by giving a set of hypotheses for each time step during inferences instead of just one fixed hypothesis as in the probably approximately correct (PAC) framework. The authors give a theoretical framework (similar to the one developed in PAC) to characterize under which condition a stochastic process can be learned by a time-aware (prospective learning) model. These are explained through their mathematical formulation and simple examples that highlight the main features of a stochastic process to determine if they can be “prospectively learned” or not. Finally, the authors perform a set of experiments, as real-world cases analogies of the simple examples, and show how time-aware prospective models can learn (or not) the tasks as described by their theory.

**Strengths:**

- This manuscript gives a useful theoretical base for understanding and describing a very intuitive concept of learning considering time, instead of a static data generator distribution. The theoretical framework allows for guarantees of function approximation by a class of time-varying hypotheses, provides a solution in explicit form for a binary classification problem with Gaussian inputs, and also analyzes the complexity of learning a specific type of process (periodic process).
- Furthermore, the authors can land their theoretical work in very simple illustrative examples, making the work easier to understand. This are followed by very well executed larger scale experiment to verify if their theoretical assumptions and simple example conclusions hold for more complex settings.
- The work presented has been thoroughly studied from theoretical and experimental fronts, with a coherent narrative to explain an intuitive consideration that has been used in practice for a while in machine learning (as it is similar to RL, or meta-learning, but not the same).
- Their Appendix (FAQs) was very helpful in explaining how this framework relates to other work around meta-learning or continual learning. On this front, they also highlight how other methods or problem setups complement their introduced concept.

**Weaknesses:**

- The complexity of the theoretical framework might make it challenging for some readers to follow. Despite this, the authors do seem to try to bridge the gap by providing explanations and building intuition around the theoretical developments. The appendix is particularly helpful in this regard.
- Following the previous comment, the clarity of the experimental results (section 5) seems to fall short compared to the well-developed theory part. The results for scenario 3 require further investigation, as they seem unexpected based on the theoretical predictions in the simple examples.
- While the appendix discusses how this intuition relates to existing concepts in RL and meta-learning, the "plain English" explanation, as they refer to it, can be further improved by drawing more formal connections between their framework and these established areas (See question below).

**Questions:**

1.- In definition 2 “Weak prospective learnability”, is a stochastic process “weakly learnable” if a model with a sequence of hypotheses can perform above chance with an arbitrarily low probability? Is it enough it exists one $t$ where this is true to consider it weakly learnable?

2.- Does the concept of aliasing apply when the hypothesis class is not enough to cover a periodic stochastic process as in example 1? On the other hand, would you see some redundancy in optimal hypotheses if this family is larger than the ones needed to describe the stochastic process?

3.- In section 2.1, I found a bit confusing the distinction between MLE and time-aware MLE models. Both models use MLE to find the solution and the difference is just how the hypothesis space is constructed. If this is correct, I would recommend changing it to “time-agnostic” vs “time-aware” or something along these lines.

4.- What is the intuition for the time-aware models not being able to solve scenario 3 in the experiments (section 5)? This is quite surprising considering that transformer architectures for example are auto-regressive and that there is some temporal structure that could be learned. Could you clarify what is the task structure, and how information is being given to the model, how is trained?

5.- A model with a class of hypotheses that change over time can learn prospectively if there is some temporal structure that can be exploited, but each risk term in equation 5 is instantly dependent on the current hypothesis, so the performance in the future is not affected by the current time step hypothesis. Why then might it be beneficial to decrease the discount factor $\gamma$? if each hypothesis is only influenced by its own time step what is the problem with having a $\gamma=1$?

6.- Considering Appendix A (FAQs), I believe a clearer connection between prospective learning and related concepts could be established by explicitly comparing their optimization objectives, training procedures, and problem structures. While this would undoubtedly be a significant undertaking, given the lack of standardized descriptions in these related fields, and likely beyond the scope of this work to address every single one, it would be a valuable contribution to readers' confusion.

**Limitations:**

The authors discuss the limitations of their work.

---

> ### Author Rebuttal · Authors · 2024-08-07
>
> We thank the reviewer for their helpful comments and valuable suggestions. We are glad that they have found the theoretical and experimental fronts of our work to be thorough. We also appreciate the positive aspects of our work, they have mentioned: (1) prospective learning as a useful theoretical foundation to describe including time in learning problems, (2) simple experiments that illustrate the theoretical findings, (3) well-executed large-scale experiments verifying the theoretical claims, and (4) clear descriptions of how prospective learning is related to other learning paradigms.
>
> We have addressed the remaining comments and questions below. If our responses are satisfactory, we would be very grateful if you can champion the fresh perspective explored in our work.
>
> > **The clarity of the experimental results (section 5\) seems to fall short compared to the well-developed theory part. The results for scenario 3 require further investigation, as they seem unexpected based on the theoretical predictions in the simple examples.**
>
> We agree that some of the experiments were not sufficiently fleshed out in the original submission. Specifically, in Scenario 3, we chose a hidden Markov model for the data where the Markov chain corresponding to the data distributions had a steady-state distribution. As we discussed on Lines 125-139, we need to discount the prospective risk over time for prospective learning to be meaningful for stochastic processes which can reach a steady-state distribution. Therefore, although the original experiments seem unappealing at first because the prospective risk is trivial, the learner does converge to the Bayes risk and this actually follows the theory completely—just that the Bayes risk is trivially large for the example we chose.
>
> We have fixed this now by choosing a hierarchical hidden Markov model where the process cannot reach a steady-state distribution. Please see the update experiment in Fig. 5 in the Rebuttal PDF.
>
> > **Q1) In definition 2 “Weak prospective learnability”,.......  Is it enough it exists one time point t where this is true to consider it weakly learnable?**
>
> You are correct. We will fix the definition. For a process to be weakly-learnable, there must exist a time $t’$ such that for all times $t\>t’$, the inequality on risks holds.
>
> > **Q2) Does the concept of aliasing apply when the hypothesis class is not enough to cover a periodic stochastic process as in example 1? On the other hand, would you see some redundancy in optimal hypotheses if this family is larger than the ones needed to describe the stochastic process?**
>
> Great question\! When the hypothesis space is not rich enough, i.e., there does not exist a hypothesis whose prospective risk matches the Bayes risk, then time-aware ERM learns the optimal hypothesis in the hypothesis class (best-in-class). This can lead to aliasing. For example, if data is drawn from a periodic process with period 5 and the hypothesis only contains sequences with period at most 3, then the prospective risk of even the best in class hypothesis can be trivial due to aliasing. On the other hand, if the hypothesis class is too large for the uniform concentration assumption in Theorem 1 to hold, then the risk of time-aware ERM might not converge to Bayes risk.
>
> The above paragraph is about hypothesis-class-based aliasing. There is another form of aliasing that can occur, this is due to sampling. Suppose we have a periodic process, but the samples we observe from it are received at a rate lower than the Nyquist rate. Models trained on such samples will suffer from aliasing. Sampling-based aliasing can also occur if the resolution of the time encoding is not sufficient. In simpler words, if the time instant of each datum is not recorded precisely, then the time-aware ERM learner might not see high-frequency fluctuations in the data stream. This is similar to how audio quality drops if an MP3 file is sub-sampled below 44.1 kHz.
>
> In general, the answer to your question is similar to the corresponding answer one might give for PAC learning. We need some inductive bias for learning to be possible. This inductive bias can come from guessing the period (either directly, or searching for the correct period like we do in structured risk minimization), or from guessing the required resolution of the time encoding (one might need to record time in seconds for high-frequency data, but only in weeks for visual recognition data).
>
> > **Q3) Both models use MLE to find the solution and the difference is just how the hypothesis space is constructed. If this is correct, I would recommend changing it to “time-agnostic” vs “time-aware” or something along these lines.**
>
> We agree that it is confusing. We will fix this to say “time-agnostic ERM” vs. “Prospective ERM”.
>
> > **Q4) What is the intuition for the time-aware models not being able to solve scenario 3 in the experiments (section 5)? This is quite surprising considering that transformer architectures for example are auto-regressive and that there is some temporal structure that could be learned.**
>
> In Scenario 3, we had chosen a hidden Markov model for the data where the Markov chain corresponding to the data distributions had a steady-state distribution. As we discussed on Lines 125-139, we need to discount the prospective risk over time for prospective learning to be meaningful for stochastic processes which can reach a steady-state distribution. The prospective risk does converge to the Bayes risk—just that the Bayes risk is trivially large for the example we chose.
>
> Please see the updated experiment in Fig. 5 in the Rebuttal PDF.
>
> To clarify the experimental setup in Scenario 3, we generated hidden states of an HMM using a transition matrix $\\Gamma\_4$ on Line 740 in Appendix C. Each hidden state corresponds to a specific distribution/task, in our case, a set of classes, detailed on Line 728 in Appendix C. Data are drawn from this distribution/task.

---

> ### Author Response · Authors · 2024-08-07
> **Rebuttal by Authors pt. 2**
>
> > **Q5) A model with a class of hypotheses that change over time can learn prospectively if there is some temporal structure that can be exploited, but each risk term in equation 5 is instantly dependent on the current hypothesis, so the performance in the future is not affected by the current time step hypothesis. Why then might it be beneficial to decrease the discount factor gamma \= 1 if each hypothesis is only influenced by its own time step what is the problem with having a gamma=1**
>
> This is a good point. Yes, future loss at time $t’$ is not affected by the current hypothesis at time $t$. We choose the discounting factor to emphasize some of the real world scenarios that only want to predict the near future. In Scenario 3, for a mixing Markov chain, choosing $\\gamma=1$ can give trivial solutions; after the Markov chain mixes completely, there are no long-term predictable patterns. However, in the near future when the chain has not mixed completely, time-aware ERM can exploit the correlations for prospective learning. One can restrict the time-horizon in prospective learning by simply cutting off time at some large value, or by discounting the loss.
>
> > **Q6) Considering Appendix A (FAQs), I believe a clearer connection between prospective learning and related concepts could be established by explicitly comparing their optimization objectives, training procedures, and problem structures. While this would undoubtedly be a significant undertaking, given the lack of standardized descriptions in these related fields, and likely beyond the scope of this work to address every single one, it would be a valuable contribution to readers' confusion.**
>
> > **While the appendix discusses how this intuition relates to existing concepts in RL and meta-learning, the "plain English" explanation, as they refer to it, can be further improved by drawing more formal connections between their framework and these established areas (See question below).**
>
> This is a very good idea and we are very thankful for it. We will update Appendix A in the camera ready version with formal connections to existing ideas in the field. This will allow us to explicitly compare the different problem setups. If this content expands beyond the scope of this present paper on prospective learning, we are very keen on writing a separate manuscript. This can be very valuable to our field.

---

> > ### Comment · Reviewer_Zv5g · 2024-08-09
> >
> > Thanks to the authors for addressing my concerns. I'm generally satisfied with the clarifications provided, but I do have a few additional comments:
> >
> > In Q4: What is the difference between the original scenario 3 experiment in Section 5 and the new one? The old data distribution had a steady-state distribution, which had a higher Bayes risk—why is this? Is it because the hidden state defining the task distribution was changing constantly, but the overall sampling process was in steady-state? Is the new configuration and the results shown in Fig. 5 designed to create a significant period within training (10 timesteps) with a specific hidden state, and then clearly switch between hidden states to generate a meaningful learning signal? I did my best to rephrase what I understood from the rebuttal—is the explanation along these lines?
> >
> > Would the time-agnostic MLP model be able to solve the task if some context information (e.g., the current task ID) were given as input? Are you providing any context information to the time-aware model? If not, is the time-aware model somehow inferring the hidden state of the data generation process? If so, that would be useful to highlight as an advantage of the proposed method.
> >
> > The fact that the originally presented model was not able to learn the task properly based on the structure of the task is not a flaw in itself. In fact, keeping that experiment and using the new one to clearly point out why we see a different result now would be really helpful for the reader.
> >
> > Also, it seems that the labels in Fig. 5 are swapped.
> >
> > Q5: Thanks, this makes sense now, and I guess it is related to the explanation in Q4?

---

> > > ### Author Response · Authors · 2024-08-10
> > >
> > > Thank you for taking the time to respond to our rebuttal. We are glad that your concerns were addressed. If you are satisfied with our response, we would be grateful if you could raise your score and champion the acceptance of our paper.
> > >
> > > > **The old data distribution had a steady-state distribution, which had a higher Bayes risk—why is this?**
> > > > **What is the difference between the original scenario 3 experiment in Section 5 and the new one?**
> > >
> > > Consider the following example. Suppose we have a Markov chain corresponding to switching between two tasks $P\_1$ and $P\_2$ with a transition matrix \[\[0.2, 0.8\], \[0.8, 0.2\]\], i.e., if the data at the current timestep was drawn from $P\_1$, in the next timestep, data will be drawn from the task $P\_2$ with probability 0.8. The steady-state distribution of this Markov chain is \[0.5, 0.5\]. In other words, as the Markov chain approaches the steady-state distribution (which happens asymptotically), the prospective learner loses the ability to predict which state the Markov chain will be at some future time $t’$. The prospective risk in Eqn. (1) takes the limit $\\tau \\to \\infty$.
> > >
> > > Suppose now that task $P\_1$ has classes $\\{1, 2\\}$ and task $P\_2$ has classes $\\{2, 1\\}$, i.e., the labels are flipped, then the prospective Bayes risk is trivially 0.5. If the classes do not “clash”, i.e., $P\_1 \\equiv \\{1, 2\\}$ and $P\_2 \\equiv \\{3, 4\\}$, then notice that a learner that predicts both $\\{1, 3\\}$ for inputs corresponding to ground-truth classes 1 and 3, and both $\\{2, 4\\}$ for inputs from ground-truth classes 2 and 4 will achieve zero prospective risk. Therefore, a trivial Bayes risk in prospective can come from (a) the Markov chain having a steady-state distribution, and (b) clashes of the classes between the tasks.
> > >
> > > In the Scenario 3 example in the original submission, we had chosen an example where both (a) and (b) were occurring. The stationary distribution for Figure 3 was a weighted mixture of the distributions of the 4 tasks created from the CIFAR10 dataset (transition matrix is $\\Gamma\_4$ on Line 740 in Appendix C) and tasks were the ones given on Line 728 in Appendix C (which do “clash”). As a result, no single hypothesis can simultaneously achieve low risk on data from all 4 tasks. Hence, no hypothesis can achieve a low risk on the stationary distribution, in other words the prospective Bayes risk is high.
> > >
> > > On the other hand, the example in the rebuttal PDF considers a hierarchical HMM which does not have a stationary distribution. The hierarchical HMM transition every 10 steps across two sets of communicating states. As a result, the distribution in the future is predictable, and there exists a hypothesis that achieves a low prospective risk on this distribution.
> > >
> > > > **Would the time-agnostic MLP model be able to solve the task if some context information (e.g., the current task ID) were given as input?**
> > >
> > > Yes. However, task boundaries are not available in prospective learning.
> > >
> > > > **Are you providing any context information to the time-aware model?**
> > >
> > > No. There is no contextual information available to the time-aware learner.
> > >
> > > > **If not, is the time-aware model somehow inferring the hidden state of the data generation process?**
> > >
> > > Yes. Think of how an ideal prospective learner might work: it would observe the data at each time-step, use an algorithm like Baum-Welch to estimate the transition matrix for the tasks and build a hypothesis for each unique task. For inference at time $t’$, it would predict the distribution over tasks at time $t’$ based on the data observed up to time $t$, and use a hypothesis corresponding to the most likely task, or sample the hypothesis with a probability proportional to its corresponding task being present at time $t’$. This is of course the ideal prospective learner, and it would converge to the Bayes risk under standard assumptions (e.g., consistency of the Baum-Welch estimates).
> > >
> > > It is interesting that in Theorem 1, we could show that time-aware ERM can also converge to the Bayes risk. This is precisely because it is doing something equivalent to inferring the hidden state of the data generating process.
> > >
> > > We will include this discussion as a remark at the end of Section 4\.
> > >
> > > > **keeping that experiment and using the new one to clearly point out why we see a different result now would be really helpful for the reader.**
> > >
> > > We will keep the original example of Scenario 3 and contrast it with the new example. We agree that this should be very useful to the reader.
> > >
> > > > **Also, it seems that the labels in Fig. 5 are swapped.**
> > >
> > > Indeed. We apologize for the oversight. We will fix it.
> > >
> > > > **Q5: Thanks, this makes sense now, and I guess it is related to the explanation in Q4?**
> > >
> > > Yes, discounting the risk changes the problem to effectively have a finite horizon. If the Markov chain does not mix by the end of this effective time-horizon, the prospective Bayes risk can be non-trivial even if the tasks clash.

---

> > > > ### Comment · Reviewer_Zv5g · 2024-08-10
> > > >
> > > > Thanks to the authors for the clear responses to my concerns. I've decided to raise my score and confidence. Nice work!

---

### Official Review · Reviewer_wUXj · 2024-07-10

**Soundness:** 3
**Presentation:** 3
**Contribution:** 2
**Rating:** 6
**Confidence:** 3

**Summary:**

The paper focuses on new paradigm of learning called "prospective learning" oppose to Probably Approximately Correct (PAC) paradigm which is how current AI systems are being designed. PAC is time agnostic given the data while PL is time-aware. The paper clearly outlines different scenarios of PL with examples and distinguishes the proposed paradigm with others. Also, showcases experimental validation of PL in different scenarios.

**Strengths:**

- The paper is clearly written and easy to follow
- Paper seems to pose a new realistic paradigm of learning which is incorporates time components compared to traditional paradigm which is time-agnostic.
- The paper has strong theoretical background.

**Weaknesses:**

- Paper fails to explain how training on unlabeled data would be yield in the proposed paradigm.
- The authors also don't consider the real world natural scenario of non-IID data where the data is a continuous video whose stochastic process will be unknown.
- Paper lacks explanation on how PL would handle or prevent catastrophic forgetting since it is arises from the distribution shift with time in PAC paradigm.

**Questions:**

- If Data-incremental learning is more closer to the problem formulation of PL, then how does multiple epochs are handled across all three cases of data type (IID, non-IID, etc)?

**Limitations:**

Yes

---

> ### Author Rebuttal · Authors · 2024-08-07
>
> We thank the reviewer for their comments on our work. We are glad that they recognize our theoretical contributions.
>
> > **Paper fails to explain how training on unlabeled data would be yield in the proposed paradigm.**
>
> This is a good question. We have been inspired by the work of Steve Hanneke \[1\]. They develop some results for the so-called “self-adaptive setting” where in addition the past data ($z\_{\\leq t}$ in our notation), the learner also has access to future inputs $\\{x\_s: t \\leq s \\leq t’\\}$ when it makes a prediction at time $t’$. Hanneke showed that there exist optimistically universal learners for the self-adaptive setting. In simple words, if a stochastic process can be self-adaptively learned strongly, then there exists a learner which does so. Just like our Theorem 1, their result is a consistency result.
>
> Hanneke does not consider the setting where the true hypothesis changes over time, which is what we are interested in. But we believe that their work gives us a nice foundation to build upon as we try to understand how to use unlabeled future data for prospective learning. This is part of future work. We will mention this in the discussion.
>
> \[1\] Hanneke, Steve. "Learning whenever learning is possible: Universal learning under general stochastic processes." *Journal of Machine Learning Research* 22.130 (2021): 1-116.
>
> > **The authors also don't consider the real world natural scenario of non-IID data where the data is a continuous video whose stochastic process will be unknown.**
>
> In fact, prospective learning is ideally suited to address problems with non-IID data, e.g., situations where one is modeling data from a video. The reason for this is as follows.
>
> Theorem 1 works for general stochastic processes. It does not need to know the class of the stochastic process, e.g., whether it is a Markov chain, or a hidden Markov model etc. This is why we think it is quite remarkable. Theorem 1 says that in spite of such a general setting, given samples from the process up to time $t$, we can build a sequence of hypotheses that can be used for any time in the future. Just like standard empirical risk minimization (ERM) allows one to build a hypothesis that can be used for any test datum drawn from the same distribution, Theorem 1 allows us to learn any stochastic process using a procedure that is conceptually quite similar to ERM, except that the predictor takes time as input. The consistency of standard ERM is one of the first results of PAC learning—and a cornerstone result. Theorem 1 is a similar result for prospective learning. Just like standard ERM makes certain assumptions about the concentration of the empirical loss and the learner’s hypothesis class, Theorem 1 also makes assumptions about consistency and concentration in Eqns. (3-4). These are rather standard, and quite benign, assumptions.
>
> Real-world stochastic processes, e.g., images in a video, may or may not satisfy the conditions of Theorem 1\. This is no different from how real-world data may or may not satisfy the assumptions of standard PAC learning. We cannot easily verify the assumptions of PAC learning in practice (whether the hypothesis class contains a model which achieves zero population loss equal to Bayes error, and whether we have uniform concentration of the empirical loss to the test loss). And similarly, assumptions of prospective learning may be difficult to verify in practice. But this does not stop us from implementing PAC learning. And similarly, one can implement prospective learning for video data.
>
> The most important practical recommendation of our paper for implementations on video data is that the architecture should encode the time of each image frame, i.e., implement time-aware ERM in Eqn. (5).
>
> > **Paper lacks explanation on how PL would handle or prevent catastrophic forgetting since it is arises from the distribution shift with time in PAC paradigm.**
>
> Catastrophic forgetting in PAC learning occurs when future training data has a different distribution than past data. This is exactly the setting addressed in prospective learning. In simple words, the central idea of our paper is that if the distribution of data changes in a predictable fashion, then prospective learning can prevent catastrophic forgetting. To give an example, if the stochastic process is periodic, i.e., there is a finite number of distinct distributions that are seen as a function of time, then time-aware ERM in Theorem 1 selects a sequence of hypotheses, one element is assigned to each of these marginal distributions. And given a test datum at a particular future time, the learner simply selects the appropriate predictor for that time. Standard PAC learning would not be able to address such periodic shifts because it does not model changes over time.
>
> > **If Data-incremental learning is more closer to the problem formulation of PL, then how does multiple epochs are handled across all three cases of data type (IID, non-IID, etc)?**
>
> Prospective learning enforces no computational constraints on the learner. This is different from other settings where one might be interested in such constraints, e.g., for computational reasons in data incremental learning, or for biological reasons in continual learning. A prospective learner is allowed to train on all past data for as many epochs as necessary. To clarify, as mentioned in footnote 7 on Page 8, we do not do any continual updates for any of the experiments.

---

> > ### Comment · Reviewer_wUXj · 2024-08-12
> >
> > Thanks to the authors for addressing my concerns. I am satisfied with the above clarifications.

---

### Official Review · Reviewer_2cje · 2024-07-11

**Soundness:** 3
**Presentation:** 3
**Contribution:** 3
**Rating:** 5
**Confidence:** 2

**Summary:**

Update: I read the rebuttal and I found it convincing, especially the explanation of the main theorem of the paper.  Additionally, the time-aware ERM idea based on time-conditioning sounds nice.
---
This paper proposes a prospective learning framework in which a sequence of future hypotheses is produced using all examples which have already been observed over the course of training.  In principle, this framework allows for problems where the data is drawn from a stochastic process which introduces temporal correlations and non-identical distributions over time steps.  The theory introduced for this seems interesting, but I don't see any particularly surprising results.  In particular, I don't see any constructive results establishing a benefit for using the "prospective transformer" as opposed to the other methods, such as conditioning on the time step.  Finally, the experimental setups seem very simplistic and contrived, and the conclusions of the experiments seem muddled.

notes from reading the paper:
  -Prospective learning is time-aware alternative to PAC learning.
  -Paper shows failure cases of PAC learning on some simple problems and also establishes some new algorithms on MNIST/CIFAR variants.
  -Prospective learning assumes data is drawn from an unknown stochastic process.
  -Data defined as z_t = (x_t, y_t).
  -Prospective learner maps from history of data to sequence of hypotheses on all time steps.
  -Only focus on expected future loss, and not risk.

**Strengths:**

-The paper is very nicely written and the idea is presented cleanly and clearly.

  -The topic of improving learning in non-stationary environments and framing this problem well, is very important.

**Weaknesses:**

-The experimental settings seem very artificial.  Additionally, the difference between the methods is unclear in terms of performance, and none of them generally achieve the bayes optimal performance.

**Questions:**

-Do you think it could be possible to use reinforcement learning (games) to get a more natural application for your framework?  For example, in a game like montezuma's revenge, the agent should get farther over the course of training and will be able to see new challenges and new parts of the environment.

**Limitations:**

-The proposed "prospective transformer" I believe is not new, but was explored here (and I believe in many earlier papers as well), yet I don't see that these were cited (https://arxiv.org/pdf/2404.19737).

---

> ### Author Rebuttal · Authors · 2024-08-07
>
> We thank the reviewer for their comments. We are glad that they find our theoretical contributions to be important for learning in non-stationary environments. We believe we have addressed all your concerns in the response below. If you think these responses are satisfactory, we would be very grateful if you can update your score.
>
> > **The theory introduced for this seems interesting, but I don't see any particularly surprising results. In particular, I don't see any constructive results establishing a benefit for using the "prospective transformer" as opposed to the other methods, such as conditioning on the time step**
>
> Let us argue why Theorem 1 is somewhat remarkable. Consider a general stochastic process (does not have to be periodic, or reach a steady-state distribution). Theorem 1 says that in spite of such a general setting, given samples from the process up to time $t$, we can build a sequence of hypotheses that can be used for any time in the future. Just like standard empirical risk minimization (ERM) allows one to build a hypothesis that can be used for any test datum drawn from the same distribution, Theorem 1 allows us to learn any stochastic process using a procedure that is conceptually quite similar to ERM, except that the predictor takes time as input. The consistency of standard ERM is one of the first results of PAC learning—and a cornerstone result. Theorem 1 is a similar result for prospective learning. It is a definitive first step on this important problem.
>
> We have made some minor modifications to the experimental parts of this paper since the deadline. Please see the common response to all reviewers above.
>
> In the context of your question, we have now figured out how to exactly implement a time-aware ERM in Eqn (5). It is simply a network that is trained on a dataset of inputs (t, X\_t) to predict the outputs Y\_t. Any MLP, CNN, or an attention-based network can be repurposed to use this modified input using an encoding of time (Line 326). In particular, there is no need to use an auto-regressive loss like we had initially done in the NeurIPS submission.
>
> Theorem 1 directly suggests this practical solution. Roughly speaking, for a practitioner who wishes that their models do not degrade as data changes over time, our paper proves, theoretically and empirically, that appending time to the train and test input data is sufficient. We suspect this is a big deal.
>
> > **The experimental settings seem very artificial. Additionally, the difference between the methods is unclear in terms of performance, and none of them generally achieve the bayes optimal performance.**
>
> This is a theoretical paper. We have constructed a, more or less, exhaustive set of “scenarios” (IID data in Scenario 1, independent but not identically distributed data in Scenario 2,  neither independent nor identically distributed data in Scenario 3, and situations where past predictions affect future data in Scenario 4). Our goal while doing so is to study precisely the performance of our proposed time-aware ERM and some baseline methods. In particular, we can calculate the Bayes risk for the experiments in Fig 1\. This is the gold standard for any method.
>
> When using non-synthetic data, our experimental scenarios are quite similar to those in existing papers on continual learning. The gold standard for experiments on non-synthetic data would be Oracle, which is a learner that knows exactly the distribution from which the test datum at any future time $t’$ is sampled; the risk of Oracle is therefore even “better” than Bayes risk. As we see, the training curves in Fig. 2 do converge to the risk of Oracle over time.
>
> In Scenario 2, time-aware ERM achieves Bayes risk while time-agnostic methods only achieve trivial risk. This shows that time-agnostic methods fail to solve even a simple synthetic problem.
>
> In our original Scenario 3, the transition matrix for the Markov chain of the hidden states was chosen to be such that the chain could converge to a steady-state distribution. As we discussed on Lines 125-139, prospective learning is only meaningful in these settings if one uses a discounted loss. Our experiments were using the non-discounted loss, and therefore the prospective risk of time-aware ERM was converging to the trivial risk. Time-agnostic ERM converges to the trivial risk for Scenario 3, in general.
>
> We have rectified the situation now using a different hidden Markov chain which does not have a steady-state distribution. See Fig. 5 in the PDF Rebuttal. Time-aware ERM for the modified Scenario 3 does converge to the Bayes risk over time.
>
> > **Do you think it could be possible to use reinforcement learning (games) to get a more natural application for your framework? For example, in a game like montezuma's revenge, the agent should get farther over the course of training and will be able to see new challenges and new parts of the environment.**
>
> This is a very good idea. In our Scenario 4, we have discussed the long term risk of a Markov Decision Process. We also developed an interesting “algorithm” for this in Appendix F. In broad strokes, this setting is similar to the problem in Montezuma’s revenge where future data depends upon past decisions. We have not instantiated the learner for Scenario 4 in experiments on non-synthetic data yet. This is mostly because the current algorithm in Appendix F involves learning both the value function and forecasting the future data, this would be difficult to pull off together. We plan to investigate this in the future.

---

> ### Author Response · Authors · 2024-08-07
> **Rebuttal by Authors Pt. 2**
>
> >  **The proposed "prospective transformer" I believe is not new, but was explored here (and I believe in many earlier papers as well), yet I don't see that these were cited**
>
> We thank the reviewer for pointing us to this, and we will cite this in our paper. In the linked paper, the authors introduce an auxiliary loss during training that forces the model to predict the next $n$-tokens at once. But for inference, they only perform next-token prediction. In “our prospective transformer”, we have the network predict the outcome of a future datum, given past data in context. During inference, both networks (prospective and auto-regressive) in our work are capable of making predictions arbitrarily far into the future. This was made possible by the choice of the positional (time) embedding we use in these models, which is different from the ones used in language modeling.
>
> All this said, please see the common response to all reviewers which discusses our proposed modifications to the experimental parts of this paper. We have now figured out how to exactly implement a time-aware ERM in Eqn (5). It is simply a network that is trained on a dataset of inputs (t, X\_t) to predict the outputs Y\_t. Any MLP, CNN, or an attention-based network can be repurposed to use this modified input using an encoding of time (Line 326). There is no need to use the prospective or auto-regressive Transformer to implement prospective learning like we had initially done in the submission. We will therefore remove the method named “prospective Transformer” in the camera ready version of the paper.
>
> We will summarize this response as a footnote and cite the paper that you have linked to.

---

### Official Review · Reviewer_NX2Q · 2024-07-24

**Soundness:** 4
**Presentation:** 4
**Contribution:** 3
**Rating:** 7
**Confidence:** 4

**Summary:**

The paper develops a new theoretical framework to address machine learning problems where data distributions and objectives evolve over time. Unlike the traditional PAC learning framework that assumes static distributions, this paper introduces "Prospective Learning" (PL), which models data as a stochastic process and emphasizes minimizing future risk based on past data. This approach integrates time as a crucial element in learning goals and algorithm design.

**Strengths:**

I like the fact that the paper considers fundamental problem and very general framework, inspired by motivating examples such as biological systems. There are many solid contributions. First, the paper formally defines PL by incorporating time into the hypothesis class, distinguishing it from PAC learning, and characterizes the stochastic processes that are either strongly or weakly prospectively learnable, offering theoretical guarantees for PL under reasonable assumptions. Second, it proposes a "time-aware" version of ERM, showing that factoring in time during hypothesis selection can solve prospective learning problems that traditional ERM methods, which ignore time, cannot. Lastly, the paper includes numerical experiments on datasets like MNIST and CIFAR-10, illustrating that time-aware learners significantly outperform their time-agnostic counterparts in dynamic environments. The paper is well-written and the main messages are quite clear.

**Weaknesses:**

The theoretical results are primarily asymptotic, and the applicability to finite-sample settings requires further investigation. In the definition of the prospective risk around (1), the limit is taken to the infinity. In reinforcement learning there are both asymptotic regret with a discounted factor (which looks similar to the definition in the paper) as well as finite-sample regret. Is it possible to develop some results for finite-sample, or having to rely on asymptotic is a limitation of the current framework?

Implementing time-aware ERM and scaling it to modern large-scale ML and popular models may be challenging, particularly for deep architecture, big data, and real-time applications. The paper would benefit from a more detailed discussion on the computational complexity and practical considerations of the proposed methods. Comments on either computational complexity from the theory side or practical scalability from the application side are welcomed.

**Questions:**

When comparing with online learning and sequential decision making, the author emphasize that the optimal hypothesis can change over time. However, there has been a line of literature (e.g., see Besbes et al. 2015 and related literature) studying the so-called “dynamic regret” and non-stationary online learning, where the optimal hypothesis can change every round. This line of work proposes various conditions to characterize the learnability and complexity of non-stationary online learning, such as the variation of the loss function. I think a comparison of prospective learning and non-stationary online learning should be added into the paper.

Omar Besbes, Yonatan Gur, and Assaf Zeevi. Non-stationary stochastic optimization. Operations Research, 63(5):1227–1244, 2015.

**Limitations:**

The authors discuss the limitations of the paper while more discussion on comparison to dynamic regret and non-stationary online learning, finite-sample guarantees, and computational complexity, should be added. The paper is primarily theoretical and does not have direct societal consequence.

---

> ### Author Rebuttal · Authors · 2024-08-07
>
> We thank the reviewer for their helpful comments and valuable suggestions. We are glad that they recognize that prospective learning addresses a fundamental problem. And we are glad that they recognize the theoretical and experimental contributions of our work. If the Reviewer thinks our responses are satisfactory, we would be very grateful if they can increase their score.
>
> > **The theoretical results are primarily asymptotic, and the applicability to finite-sample settings requires further investigation. In the definition of the prospective risk around (1), the limit is taken to the infinity. In reinforcement learning there are both asymptotic regret with a discounted factor (which looks similar to the definition in the paper) as well as finite-sample regret. Is it possible to develop some results for finite-sample, or having to rely on asymptotic is a limitation of the current framework?**
>
> The result in Theorem 1 is about the asymptotics. In Remark 2, we do provide a sample complexity bound for prospective learning of periodic processes. As future work, we are indeed exploring sample complexity of prospectively learning more general stochastic processes such as Hidden Markov models.
>
> The limit as $\\tau \\to \\infty$ on Line 80 in the definition of prospective risk is different from assuming an asymptotically large number of samples in Theorem 1\. Line 80 uses the infinite horizon risk for the following reason. In prospective learning, if one uses a finite time horizon, then it becomes similar to a multi-task learning problem. This is investigated in works like \[27, 28, 29\] cited in the paper. We focussed on the infinite horizon prospective learning problem to force the learner to model the evolution of the stochastic process. We appreciate the Reviewer’s point. For some processes, e.g., Markov processes that reach a steady-state distribution, it is not possible to prospect arbitrarily far into the future. For such processes, we have to use a discounted prospective risk (mentioned on Line 82). We have now proved a corollary of Theorem 1 for discounted risks, which we will add to the main paper. We will also cite the finite-sample regret bounds from the RL literature.
>
> > **Implementing time-aware ERM and scaling it to modern large-scale ML and popular models may be challenging, particularly for deep architecture, big data, and real-time applications. The paper would benefit from a more detailed discussion on the computational complexity and practical considerations of the proposed methods. Comments on either computational complexity from the theory side or practical scalability from the application side are welcomed.**
>
> We have figured out a few minor changes to the experimental section of the paper since the deadline. Please see the common response to all Reviewers for these proposed changes.
>
> Time-aware ERM is actually quite easy to implement. It is simply a network that is trained on a dataset of inputs (t, X\_t) to predict the outputs Y\_t. Any MLP, CNN, or an attention-based network can be repurposed to use this modified input using an encoding of time (Line 326). In practice, perhaps the only thing one must be careful about is that the encoding of time should be sufficiently rich to incorporate all past data. In other words, we are interested in an absolute encoding of time, unlike Transformers where position encoding is used only within the context window. This is easy to achieve using a large number of logarithmically-spaced Fourier frequencies, or a binary encoding time in minutes/seconds, like we have shown in the experiments.
>
> We will add the above response as a remark in the paper.
>
> > **When comparing with online learning and sequential decision making, the author emphasize that the optimal hypothesis can change over time. However, there has been a line of literature (e.g., see Besbes et al. 2015 and related literature) studying the so-called “dynamic regret” and non-stationary online learning, where the optimal hypothesis can change every round. This line of work proposes various conditions to characterize the learnability and complexity of non-stationary online learning, such as the variation of the loss function. I think a comparison of prospective learning and non-stationary online learning should be added into the paper.**
>
> > **Omar Besbes, Yonatan Gur, and Assaf Zeevi. Non-stationary stochastic optimization. Operations Research, 63(5):1227–1244, 2015\.**
>
> Thank you for the reference. We will clarify the distinction in the camera ready version.
>
> Although in both cases the optimal hypothesis changes over time, the stochastic approximation problem studied in Besbes et al. 2015 and the prospective learning problem studied in this paper are somewhat different. The former optimizes over stochastic processes X\_t to minimize a dynamical regret, finding the stochastic approximation of the optimal point of the regret, while the latter optimizes over hypothesis class H\_t to find the best hypothesis.
>
> > **More discussion on comparison to dynamic regret and non-stationary online learning, finite-sample guarantees, and computational complexity, should be added**
>
> We appreciate the reviewer’s constructive feedback and we will expand the paper to include a detailed discussion on the topics above. Our plan is to expand the FAQ to discuss these ideas and formally compare prospective learning to these ideas.

---

> > ### Comment · Reviewer_NX2Q · 2024-08-11
> >
> > Thank you for your detailed answer to my questions! I am satisfied with the feedback and maintain my score, in favor of accepting the paper.

---

### Author Rebuttal · Authors · 2024-08-07

### **Common response to all Reviewers**

We thank the reviewers for lending their expertise to assess our work and for helping us improve it. We are glad that the reviewers are positively inclined towards this work. The find that the problem is fundamental, and our paper presents a general framework with strong theory [NX2Q, wUXj, Zv5g], the problem is important, the paper is well-written [2cje, wUXj], the work is thorough with illustrative examples [Zv5g]. Individual responses to each reviewer are included below.

Based on Reviewer feedback, we will make the following changes to the manuscript to improve the experimental parts of the paper.

**New experiments on Scenario 3 (neither independent nor identically distributed data).**
For Scenario 3, we had chosen a Hidden Markov Model where the Markov chain corresponding to the data distribution had a steady-state distribution. As we discussed on Lines 125-139, one needs to discount the prospective risk over time for learning to be meaningful for such processes. Experiments on CIFAR-10 in Fig. 3 did not use discounting and that is why the prospective risk was high. The experiment was correct, just that even the Bayes risk for this process is quite large (see Lines 125-139).

In Fig 5 in the Rebuttal PDF, using synthetic data, we set up a hierarchical HMM where the underlying Markov chain has two communicating sets of states, and the chain transitions across these sets deterministically after 10 timesteps. This process does not have a steady state distribution. And like our theory suggests, time-aware ERM (renamed to Prospective ERM) converges to the Bayes risk. We will add this figure to the paper. We will also conduct a similar experiment using a hierarchical HMM on CIFAR-10 data to replace Fig. 3; we expect the results to be similar to those in Fig. 5.

**Comparing time-aware ERM with other continual learning algorithms.**
Prospective learning is designed to address situations when data, and the optimal hypothesis, evolves over time. Task-agnostic online continual learning methods are the closest algorithms in the literature that can work in this setting. We implemented (a) Follow The Leader, (b) Online SGD, and (c) online variational Bayes [1]. These algorithms are not designed for prospective learning but they are designed to address the changing data distribution $t$. The setup is identical to that of Fig. 2 in the paper, i.e., Scenario 2 with independent but not identically distributed data. As Fig. 4 in the Rebuttal PDF shows, on tasks constructed from both synthetic and MNIST data, these baseline algorithms achieve trivial prospective risk. Even the average risk on *past data* for these baselines is trivial (0.5 for synthetic data, 0.75 for MNIST). This experiment suggests that these three methods cannot handle changing data distributions. Even if there are only two distinct distributions of data, and these changes are predictable. In contrast, the prospective risk of Time-MLP converges to zero over time.

Fig. 4 in Rebuttal PDF will replace Fig. 2a and 2b in the current manuscript. We will conduct a similar experiment on CIFAR-10 to replace Fig. 2c.

See footnote 7 on Page 8 in the paper. In our experiments, for every time $t$ we use 3 training datasets that are realizations of $z\_{\\leq t}$ and calculate the prospective risk on 100 realizations of $Z\_{\> t}$. For a time-horizon of 400, this entails 1,200 trained models for each method. This is why we could not finish similar experiments on CIFAR-10. But we will add them to the camera ready.

[1] Zeno, Chen, et al. Task agnostic continual learning using online variational Bayes. arXiv:1803.10123 (2018).

**Cleaning up the different learners. There will be only two methods: Time-agnostic and Time-aware ERM (renamed to Prospective ERM); Auto-regressive and prospective Transformer will be moved to the Appendix.**
We have now figured out a way to exactly implement a time-aware ERM (will be renamed to Prospective ERM) in Eqn (5). Theorem 1 suggests that we must simply use a network that is trained on a dataset of inputs $(t, X\_t)$ to predict the outputs $Y\_t$. Any MLP, CNN, or an attention-based network can be repurposed to use this modified input using an encoding of time (Line 326). In particular, there is no need to use an auto-regressive loss, or fit a prospective Transformer like we had initially done in the manuscript. Roughly speaking, for a practitioner who wishes that their models do not degrade as data changes over time, our paper proves, theoretically and empirically, that appending time to the train and test input data is sufficient.

We will therefore move experiments that use auto-regressive and prospective Transformers in Fig. 2 and 3 to the Appendix. Continual learning baselines will be added to those figures instead, as discussed above.

**Large language models may not be good prospective learners**
It is an interesting question as to whether LLMs which are trained using auto-regressive likelihoods with Transformer-based architectures can do prospective learning. To study this, we used LLama-7B and Gemma-7B to evaluate the prospective risk for Scenarios 1 to 3\. The prompt contains a few samples from the stochastic process (sub-sequences of $(Yt)$ consisting of 0s and 1s) and an English language description of the family of stochastic processes that generated the data. The LLM is tasked with completing the prompt with the next 20 most likely samples. The data and evaluation setup is identical to Fig. 1 in the paper. As Fig. 6 in the Rebuttal PDF shows, prospective risk does not improve over time for any of the LLMs. In contrast, risk of the time-aware MLE in Fig. 1 in the main paper or Fig. 6 in the Rebuttal converges to the Bayes risk. These results will be added to Section 5.

We believe these changes to the experimental section will improve the quality of our paper. We are very thankful to the Reviewers for suggesting some of these changes.

---

> ### Author Response · Authors · 2024-08-10
> **Gentle reminder**
>
> We thank the reviewers for their feedback. We eagerly await your response to our rebuttal and would love an opportunity to discuss your concerns. If you think our rebuttal is satisfactory, we would be grateful if you could increase your score.

---

### Decision · Program_Chairs · 2024-09-25

**Decision:**

Accept (poster)

**Comment:**

This paper introduces "prospective learning" as a theoretical framework to address distributions evolving over time following a stochastic processes. The authors quantify processes that are prospectively learnable and prove time-aware ERM can solve them. This work takes an interesting step toward ML systems that incorporate time-varying distributions.

The reviewing team has raised several important feedback, including i) clarifying connections to non-stationary online learning, continual learning, RL, and meta-learning and ii) improving the clarity of the experimental results and adding additional baselines such as continual learning methods. The paper will be significantly strengthened by incorporating them in the camera-ready version.